# Valorisation of CO₂ into Value-Added Products via Microbial Electrosynthesis (MES) and Electro-Fermentation Technology

**Marzuqa Quraishi** [1,†], **Kayinath Wani** [2], **Soumya Pandit** [2,*], **Piyush Kumar Gupta** [2],
**Ashutosh Kumar Rai** [3,†], **Dibyajit Lahiri** [4], **Dipak A. Jadhav** [5], **Rina Rani Ray** [6], **Sokhee P. Jung** [7],
**Vijay Kumar Thakur** [8,9,10] **and Ram Prasad** [11,*]

1  Amity Institute of Biotechnology, Amity University, Mumbai 410206, India; marzuqa20@gmail.com
2  Department of Life Sciences, School of Basic Sciences and Research, Sharda University,
   Greater Noida 201310, India; kainaatwani38@gmail.com (K.W.); dr.piyushkgupta@gmail.com (P.K.G.)
3  Department of Biochemistry, College of Medicine, Imam Abdulrahman Bin Faisal University,
   Dammam 31441, Saudi Arabia; akraibiotech@gmail.com
4  Department of Biotechnology, University of Engineering & Management, Kolkata 700156, India;
   dibyajit.lahiri@uem.edu.in
5  Department of Agricultural Engineering, Maharashtra Institute of Technology, Aurangabad 431010, India;
   deepak.jadhav1795@gmail.com
6  Department of Biotechnology, Maulana Abul Kalam Azad University of Technology, Kolkata 700064, India;
   raypumicro@gmail.com
7  Department of Environment and Energy Engineering, Chonnam National University, Gwangju 61186, Korea;
   sokheejung@chonnam.ac.kr
8  Biorefining and Advanced Materials Research Centre, Scotland's Rural College, Edinburgh EH9 3JG, UK;
   Vijay.Thakur@sruc.ac.uk
9  School of Engineering, University of Petroleum & Energy Studies (UPES), Dehradun 248007, India
10 Department of Mechanical Engineering, School of Engineering, Shiv Nadar University, Noida 201314, India
11 Department of Botany, Mahatma Gandhi Central University, Motahari 845401, India
*  Correspondence: sounip@gmail.com (S.P.); rpjnu2001@gmail.com (R.P.)
†  These authors contributed equally to this work.

**Abstract:** Microbial electrocatalysis reckons on microbes as catalysts for reactions occurring at electrodes. Microbial fuel cells and microbial electrolysis cells are well-known in this context; both prefer the oxidation of organic and inorganic matter for producing electricity. Notably, the synthesis of high energy-density chemicals (fuels) or their precursors by microorganisms using bio-cathode to yield electrical energy is called Microbial Electrosynthesis (MES), giving an exceptionally appealing novel way for producing beneficial products from electricity and wastewater. This review accentuates the concept, importance and opportunities of MES, as an emerging discipline at the nexus of microbiology and electrochemistry. Production of organic compounds from MES is considered as an effective technique for the generation of various beneficial reduced end-products (like acetate and butyrate) as well as in reducing the load of CO₂ from the atmosphere to mitigate the harmful effect of greenhouse gases in global warming. Although MES is still an emerging technology, this method is not thoroughly known. The authors have focused on MES, as it is the next transformative, viable alternative technology to decrease the repercussions of surplus carbon dioxide in the environment along with conserving energy.

**Keywords:** bioelectrochemical system (BES); carbon dioxide sequestration; extracellular electron transfer (EET); electroactive microorganisms; microbial biocatalyst; electro-fermentation; circular economy; downstream processing (DSP); gene manipulation

## 1. Introduction

Carbon dioxide is naturally abundant (about 0.03% to 0.04%) in the atmosphere and is eventually responsible for the ecological balance of the ecosystem [1,2]. However, the ever-increasing population and the energy demands have led to changes in the natural

cycles of greenhouse gases (including $CO_2$). Industrial emissions and the misuse of fossil fuels have led to about a 40% upsurge in the total atmospheric $CO_2$ and about a 78% rise in the greenhouse gases concentration from 1990 to 2016. Hence, the accumulation of carbon dioxide has led to absorption and re-emission of heat, attributing to an additional warming of the planet [2–7]. The changes in the land-use practices predominantly, deforestation and more use of agricultural land, cement production, use of fossil fuels for energy generation and transportation are the major factors contributing to the carbon emissions [3–6]. $CO_2$ can be captured and converted into carbon-neutral value-added products via microbial electrosynthesis (MES) [8].

MES is a novel microbial electrochemical technology that supplies electrons to microorganisms via an electric current (biocathode-driven i.e., biofilm + cathode) inside an electrochemical cell. These microbes act as biocatalysts and use the electrons for reducing carbon dioxide to eventually yield industrially relevant products like transportation fuels [9,10]. It is a fascinating alternative for capturing and expanding the value of the electrical energy generated from recurrent renewable sources (like sun, geothermal, biomass, or wind) [8,11,12]. This interchange of energy to different usable carbon materials is the most riveting way for storing energy, its distribution and utilisation [10,13,14]. Production of organic compounds from MES is considered to be an effective technique for the inception of various beneficial multi-carbon reduced end-products like acetate and butyrate by the valorisation of low-value $CO_2$. Further, bio-production dependent on $CO_2$ is advantageous, as it uses less arable land and freshwater resources, has low $CO_2$ emissions, no major nutritional supplementation is needed, has excess substrate availability and lastly, chemical bonds can be employed for the storage of excess electrical energy [8,15,16]. Although MES technology is still in its infancy, it has been demonstrated as a promising green alternative for $CO_2$ sequestration and bioelectrosynthesis of high-valued multi-carbon organic compounds. The paucity of knowledge must be resolved before the commercialisation of MES technology [7,8,17–20].

The MES process imitates the natural photosynthesis process if the external power is supplied from a renewable solar source, depicting plenty of advantages as compared to the bioenergy procedure that depends on photosynthesis [9,10]. Some studies have shown that Gram-negative (most efficient being *Sporomusa ovata* DSM-2662) and Gram-positive (like *Moorella thermoacetica* and *Clostridium* spp.) acetogenic bacteria gain electrons from graphite electrodes and act as an electron ($e^-$) donor in the reduction of $CO_2$-producing multi-carbon compounds extracellularly. A strain named *Clostridium ljungdahlii* is capable of MES, which can be genetically controlled and can be used to generate high-valued commodities [21–25].

This review paper is a comprehensive analysis of numerous products obtained by the use of MES, including the downstream processing, its commercialisation potential and a few limitations. It further discusses the recent trends, emphasising MES and the role of electroactive microbes for their various applications including electricity production and wastewater treatment.

## 2. Bioelectrochemical System (BES)

Bioelectrochemical systems (BESs) are revolutionary novel bioengineering technology that has substantially diversified their scope over the past decade [26]. These are capable of converting electrical energy into chemical energy (like in microbial electrolytic cells (MECs)) and vice versa (like in microbial fuel cells (MFCs)) by degrading several organic compound substrates, especially lignocellulosic biomass derived from wastewater with the help of microbes or their enzymes to generate valuable products [2,27] such as methanol, ethanol, acetate, formate, or hydrocarbons; these commodities (being precursors) are later converted or directly used as a sustainable green alternative to fossil fuels (See Figure 1). The emerging MES process of producing high-value chemicals has greatly broadened the BES's scope. BES being an eco-friendly and energy-saving technology has gained much popularity, it revolves around $e^-$ transfer and energy transformation. Researchers are now

exploiting the design of electrochemical devices, electrodes, catalyst and separator material optimisation and screening of electroactive microorganisms [2,16,28–31].

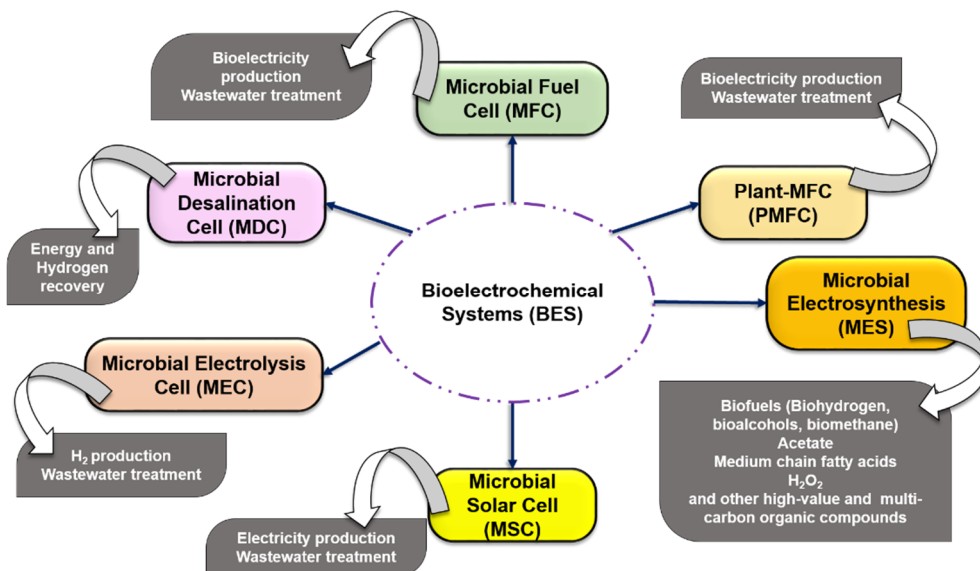

**Figure 1.** A schematic overview of multiple categories of BESs depending on the mode of their applications [28,29,31].

A typical BES consists of a cathode, anode and an optional membrane that separates the two of them. Figure 2 portrays a schematic representation of BES for $e^-$ transfer from electrodes to microorganisms. Oxidation occurs in the anodic chamber (like the oxidation of acetate or water) and the reduction takes place in the cathodic chamber (like the $O_2$ reduction or $H_2$ evolution). At least one of these two half-reactions is biocatalysed, either by microbial cells, their enzymes or their organelles. The aqueous electrolyte solution surrounds the electrodes, where the reactants and products reside [16,27–30].

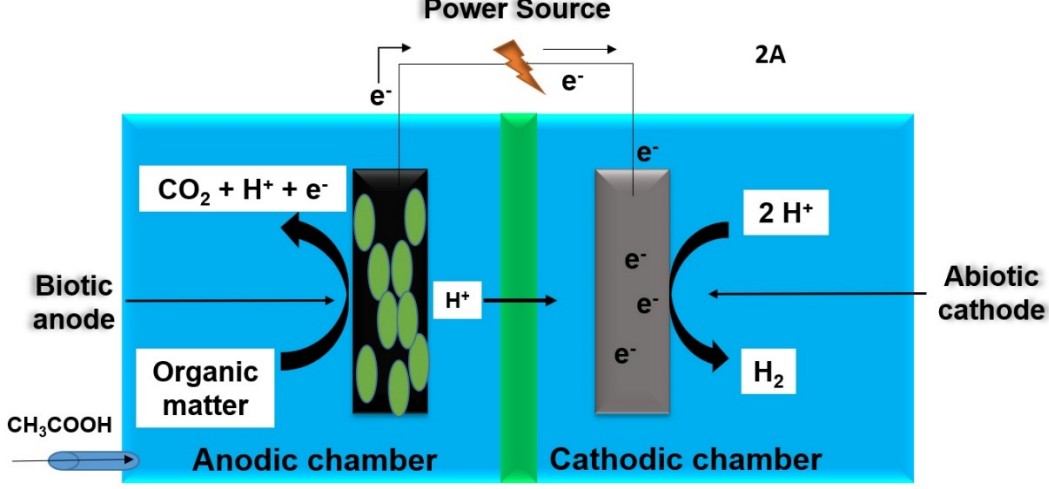

**Figure 2.** *Cont*.

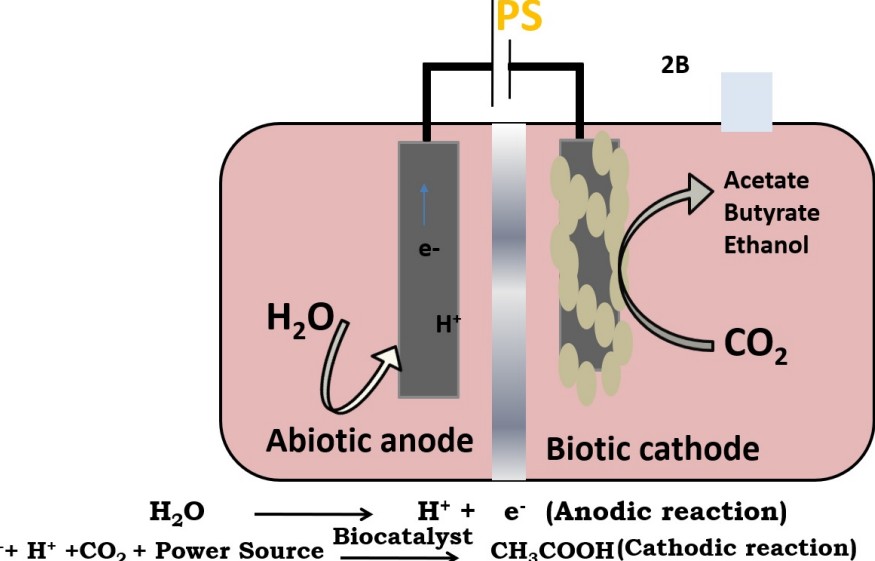

$$H_2O \longrightarrow H^+ + e^- \quad \text{(Anodic reaction)}$$
$$e^- + H^+ + CO_2 + \text{Power Source} \xrightarrow{\text{Biocatalyst}} CH_3COOH \text{(Cathodic reaction)}$$

**Figure 2. (A)** A generalised schematic representation of a typical dual-chambered Bioelectrochemical System (BES) depicting its construction and the processes carried out in the system. On the anode, microbial biofilm comprising exoelectrogens oxidises the organic matter. Electrons and protons on being released travel via distinct paths to the cathode, where they are reduced to form $H_2$. The anode potential being higher than that of the cathode ensures a non-spontaneous reaction, making it essential to apply an external power supply to facilitate the bioelectrochemical reaction. **(B)** [32–36].

For didactical reasons, the overall process has been summed up using acetate as an example, being the most prominent carbon source for MFC's bioanode during laboratory studies [37].

$$\text{Anode: } CH_3COO^- + 4H_2O_{(l)} \rightarrow 2HCO_3^- + 9H^+ + 8e^- \tag{1}$$

$$\text{Cathode: } 8\ (H^+ + e^- \rightarrow \frac{1}{2}\ H_{2(g)}) \tag{2}$$

$$\text{Overall reaction: } CH_3COO^- + 4H_2O_{(l)} \rightarrow 2HCO_3^- + H^+ + 4H_{2(g)} \tag{3}$$

When electrical power is provided to the BES system, it is said to be in Microbial Electrolysis Cell mode. The extra power is supplied to intensify the reaction kinetics and to drive thermodynamically detrimental cathodic half-reactions. The bacteria employed in an MES are typically anaerobic homoacetogenic bacteria that employ reducing agents or electrons provided by the cathode to metabolically convert $H_2$ and $CO_2$ to acetate and other chemicals. A number of lithoautotrohphs are also utilised in the metabolic conversion of $CO_2$ to acetate and other organic molecules. The Wood–Ljungdahl or acetyl-CoA route follows the anaerobic conversion to acetate Figure 2B. Optionally, MFC can be applied to deliver the power to the electrochemical circuit [16,30,38,39].

MFCs trigger the chief growth and development of the microbial electrochemistry discipline and generate electricity utilising the microorganisms that are capable of handling and growing on the electrode (in this case, anode) surface, along with the ability to use electrodes as an $e^-$ acceptor for the oxidation of organic compounds. In such systems, the electrical force is accumulated from the anodic response and the cathodic half-reactions take place simultaneously (like a decrease in $O_2$) [40].

BES system is available as planktonic microbial cells as well as a biofilm. The electroactive biofilm contains electrochemically active (EAM) and inactive microorganisms. This system having various functions like the breakdown of complex substrates proves to be beneficial. EAMs also empower the productive exchange of electrons from or towards solid-state electrodes to boost the current densities, improve the energy efficiencies and production in these systems. For the same purpose, extracellular electron transport (EET) is used to transport the $e^-$ from or towards an insoluble $e^-$ acceptor or donor (Figure 3) [41,42].

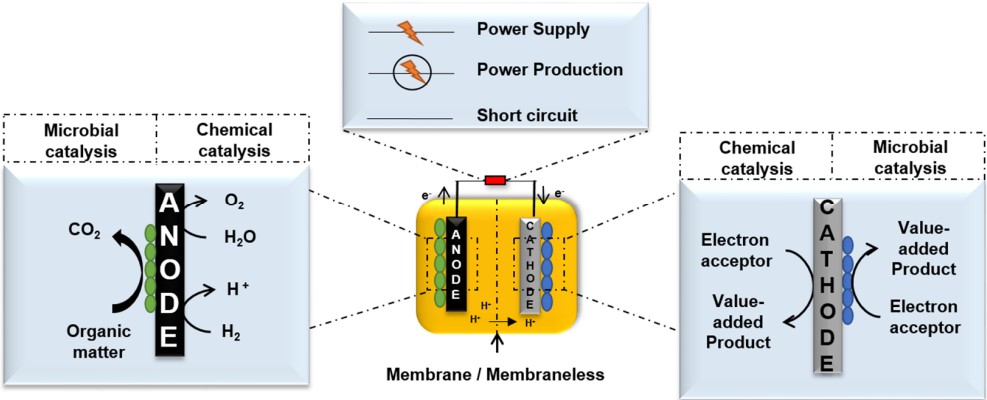

**Figure 3.** An in-depth overview of the concepts associated with the highly versatile microbe–electrode interaction-based microbial BES technology and a multitude of choices available to carry out a diverse range of processes at both anodic and cathodic chambers simultaneously [43].

### 2.1. Transmission of Electrons at the Anode

An ideal anode should have low resistance, large surface area, high electrical conductivity, anti-corrosiveness, strong mechanical strength, fouling resistance, chemical stability, good biocompatibility and scalability, preferably with ease of construction and mainly of low cost. The property of some EAM has been known over for a century to provide $e^-$ to the electrodes. However, the mechanisms of anodic EET have most extensively been investigated in the last few years. The conversion of organic substrate to electric current does not only occur in fermentation reactions, but also in natural respiratory processes. This finding was a significant advancement that aided in discovering the mechanism behind electrons being drawn from microorganisms and the existence of two diverse EET pathways. The principal pathway of direct electron transfer (DET) includes the immediate contact between the electron transfer chain (ETC) of the microorganisms and the electrode surface. The other component, mediated electron transfer, shuttles $e^-$ carriers and reversibly reduce and oxidise them in between electrodes and microbes. So far, only *Geobacter* spp., *Rhodoferax* spp. and *Shewanella* spp. have been widely researched to explore the EET mechanisms. The initial DET occurs through membrane-bound ETC proteins, like Cytochrome c (Cyt c), while the other carries $e^-$ from the microbe to the surface of the electrode along conductive pili (also known as nanowires), which are attached to membrane-bound $e^-$ transport compounds [44–46].

### 2.2. Transmission of Electrons at the Cathode

The cathode serves as a reservoir of $e^-$ donors for microbes and thus influences the potency of the process [47]. The desired cathode should have the following properties for being used as a biocathode, it should have high productivity, excellent chemical stability, biocompatibility with high mechanical strength, surface area and low cost. The most extensively studied and the most frequently used end-product of $CO_2$ conversion for cathode efficiency assessment is acetate. Carbon-based compounds in the MES frameworks are extensively employed cathodes for $CO_2$ reduction [48].

The material of the cathode plays a pivotal role in electrohydrogenesis and electromethanogenesis. The latter need less energy input as compared to the former ($-0.23$ to $-0.41$ V vs. Standard Hydrogen Electrode (SHE)). The BES with microbial biocathode relies on microbes to receive the $e^-$ from a solid electrode, acting as a donor for reducing the terminal $e^-$ acceptor, these are known as electrotrophs. Researchers have mainly focused on understanding the mechanism of the anode while information on the reverse processes like the flow of $e^-$ from electrodes to microorganisms was insubstantial. In 2004, cathodic DET flow was reported for the reduction of various forms of nitrogen (nitrate $NO_3^-$ to nitrite $NO_2^-$) from the cathode and assured at about $-0.34$ V vs. SHE, it further enriched the substances along with microbial species [18,46,49]. Several *Desulfovibrio* sp. has been

successfully employed for biohydrogen production with biocathode [50]. To date, the exact pathways of cathodic $e^-$ transfer to an electroactive acetogen are still unknown [47]. Researchers have recommended numerous mechanisms that resemble the bioanode processes but possess different redox potentials.

Some general reactions carried out by anaerobic methanogens are [51]:

$$\text{Methanol: } 4CH_3OH \rightarrow 3CH_4 + CO_2 + 2H_2O \tag{4}$$

$$\text{Hydrogen: } 4H_2 + CO_2 \rightarrow CH_4 + 2H_2O \tag{5}$$

$$\text{Metals: } 4Me_0 + 8H^+ + CO_2 \rightarrow 4Me^{++} + CH_4 + 2H_2O \tag{6}$$

$$\text{Acetate: } CH_3COOH \rightarrow CH_4 + CO_2 \tag{7}$$

$$\text{Methylamine: } 4(CH_3)NH_2 + 2H_2O \rightarrow 3CH_4 + CO_2 + 4NH_3 \tag{8}$$

Cathodic $e^-$ acceptors and their maximum power densities have been elaborated by Ucar et al. [35].

### 2.3. Electrosynthesis Assisted by Microbes

Using bioanode systems along with a chemical cathode in electrosynthesis is currently in demand [52]. $H_2$ can be generated with the help of platinum cathode MEC, the pH increases by consuming protons at the cathode. Similarly, hydrogen peroxide can be produced in BES via carbon cathode, which can later be used as a beneficial chemical. $H_2O_2$ thus produced can further be used for oxidation reactions, bioproduction and bioremediation processes [36,39,46,53–55], as well as in Fenton reaction [56]. Thus, microbial assisted electrosynthesis can efficiently be employed for the production of disinfectants or oxidants [39].

Some of the microorganisms used in MES for the production of various targets include *Sporomusa* sp., *Clostridium* sp., *Acetobacterium* sp., *Methanobacterium* sp. and many more [47]. The two main genera found to be dominant in this process include *Bacteroidetes* and *Proteobacteria*. Other than these, another prime genus of bacteria is *Firmicutes* [57]. For the production of methane, *Methanobacterium* sp. is considered to be a salient genus. Similar to the role of *Geobacter* sp. in bioanodes, these are considered vital for biocathode enrichment [58,59]. The microbes involved in the volatile fatty acid (VFA) and butyrate productions are *Megasphaera* sp. and *Clostridium* sp. In the case of hydrogen ($H_2$) production, acetogens are the superior community in the media, but $H_2$-producing bacteria control the biofilm production [60]. For the production of acetate, *Acetobacterium* sp. play a pivotal role. *Methanobacterium* sp. are also detected in acetate-producing biocathodes. Current experiments have demonstrated that microbes (predominantly *Firmicutes*) may be responsible for $H_2$ catalysis as they generate methane at low cathode potential, increasing the reducing power and production of organic compounds in MES [24,61].

Biofilm used in MES systems increases the efficiency and stability of the overall system in the long-run, by preventing the washing out of microorganisms. Nevertheless, this technology has drawbacks too, like microbe-electrode interactions and extracellular electron transfer related challenges that can be overcome by electrode engineering and optimisation [62,63]. Strategies such as modification of the electrode surface by the generation of 3D structures have been shown by Kerzenmacher [64]. Modifications such as enriching the surface of the electrode with positively charged molecules also upsurge the efficiency of the process. Alternatively, changes in the composition of the microbial community could also help. Intermixing of cultures or creating co-cultures can be used in biofilm-based MES to enhance the performance of the system as these cultures are robust to changes in the environment and are flexible with different types of substrates [65]. However, when present in abundant quantity, the species compete with one another for $e^-$, leading to a decrease in the product specificity [66]. A potential solution for the same is enriching specific species by using electrochemically-driven reactions in the long run, along with the addition of

supplements. Biofilm-based MES highlighting the different process performances have been explained by Fruehauf [65].

### 2.4. Electroactive Microbes and Extracellular Electron Transfer

Electromicrobiology explores and exploits the microbe-electron (both donor and acceptor) interaction. Currently, the use of electroactive microorganisms has become fascinating in sustainable bioengineering practices. In these electroactive microbes, $e^-$ transfer reactions encompass beyond the cell surface in a process called extracellular electron transfer (EET) [67,68], in MFCs, the electricigens (anodic catalysts) are employed. Electrogens are microbes that can release $e^-$ onto an extracellular electrode (anode) surface, resulting in a positive electric current [69]. For instance, iron-reducing exoelectrogen bacteria *Geobacter sulfurreducens* produces high power density at moderate temperatures [28,68,70–75]. On the other hand, Electrotrophs retrieve $e^-$ from an extracellular electrode (cathode) surface, resulting in an opposite (negative) electric current, like in MES [76]. For instance, Fe (II) and sulphur-oxidising bacteria *Acidithiobacillus ferrooxidans* switch their energy source from diffusible iron ions to direct $e^-$ uptake from a polarised electrode [77].

In MES, the cathodic $e^-$ autotrophic microbes convert $CO_2$ to fuels, chemicals, biodetergents, bioplastics and recover metals from metallurgy waste streams. This ability of theirs acts as an advantage for them in several environmental niches, one such distinct boon being selectively isolating rare strains, characterising them and utilising their characteristics in sustainable BES technologies [28,70,72–75]. EET characteristics and behaviour in BES of numerous organisms have been discussed by Kracke et al. [78].

Using advancing cross-disciplinary fields like material science, electrochemistry, biotechnology and MES, the ongoing energy issues can be ingeniously resolved. The main challenge in using this technology is shuttling electrons into microbes from the reductive cathode in sufficient quantities to produce products at a suitable level. This shuttling occurs in the form of redox-active compounds (e.g., the cofactor Flavin secreted by microbes), which transports $e^-$. These compounds are reduced by redox partaking enzymes such as Cyt c embedded on the surface of the microbial cell and then shuttled as electrons to the anode where they are oxidised. There are three mechanisms for the same; these are electrolysis of water, production of soluble $e^-$ mediators and direct transfer of $e^-$ (See, Table 1) [79,80].

The majority of the studies in Table 1 required a mediator to delegate microbial-anode interaction. When natural electroactive bacteria were employed as the producer strain or when the producer strain was co-cultured with an electroactive strain, an artificial $e^-$ carrier was not needed. To date, only a handful of the bacterium has been evaluated as potential aspirants for anodic electro-fermentation. Another mutual feature shared by the microbes in the table is the use of carbon-based anodes with an average 0.4 V (vs. SHE) applied potential [81].

Two prime methods for increasing the EET of microbes are the introduction of EET mechanisms from other microbes capable of MES or the overexpression of the mechanism of the gene itself. The EET chain of *Shewanella oneidensis* comprising *MtrA*, *MtrB* and *MtrC* were inserted into *E. coli* as a representative example of the previous case to create an electrical conduit on the surface of its cell. The protein inserted through the heterologous pathway are then functionally expressed in the bacterium. Further, the interaction of protein *MtrA* and the bacteria aid in accelerating the process of reduction of soluble Fe(III). The modified strains reduce the metal and solid metal ions by about 8 and 4 folds (against the wild strain) respectively [62,63,102–104] The *G. sulfurreducens* strain can overexpress in both heterologous and homologous states. Gene expression of the *pilA* gene encoding for the structure of protein pilin can be spiked by interrupting the gene that encodes for the periplasmic Cyt c. Hence, this increases the rate of iron reduction. Two genes that code for Cyt c are *GSU1771* and *GSU3274*, the second one being a more prominent target for the movement of $e^-$ [62,63,105–108].

**Table 1.** Synthesis of high-value chemicals through anodic electro-fermentation and microbial electrosynthesis (MES). Adapted from [79].

| Microbe | Substrate | Product | Mechanisms of EET | Genetic Modification of Host | Yield (Y) and/or Titre (T) | Ref. |
|---|---|---|---|---|---|---|
| **Anodic Electro-Fermentation** | | | | | | |
| *Shewanella oneidensis* | Glucose | Acetate | Direct electron transfer (ET) | Introduction of *E. coli* galactose permease (*galP*) and glucose kinase (*glk*) genes. | No | [82] |
| | Glycerol | Ethanol; Acetate | Direct ET | Introduction of *Zymomonas mobilis* ethanol production module and *E. coli* glycerol utilisation module | Y = 52% ± 4% $T = 1.28 \pm 0.02$ g L$^{-1}$; Y = 13% ± 6% $T = 0.29 \pm 0.08$ g L$^{-1}$ | [83] |
| | Lactate | Acetoin | Direct ET | Introduction of *Bacillus subtilis* acetolactate decarboxylase and acetolactate synthase; Deletion of genomic prophages; Knockout of the phosphotransacetylase and acetate kinase genes | Y = 52% $T = 0.24$ g L$^{-1}$ Productivity = 0.91 mg h$^{-1}$ | [84] |
| *Actinobacillus succinogenes* | Glycerol | Succinate; Acetate; Formate | Neutral red mediated ET | Transmembrane mediator transport was improved by atmospheric and room temperature plasma mutagenesis. | Y = 68% $T = 23.92 \pm 0.08$ g L$^{-1}$; Y = 7% $T = 1.15 \pm 0.77$ g L$^{-1}$; Y = 19% $T = 2.57 \pm 0.11$ g L$^{-1}$ | [85] |
| *Klebsiella pneumoniae* | Glycerol | Acetate; 3-Hydroxypropionic acid; 1,3-Propanediol | Direct ET | Not modified | T = 21.7 mM; T = 7.6 mM; T = 45.5 mM | [86] |
| *Clostridium cellobioparum* + *Geobacter sulfurreducens* | Glycerol | Ethanol | Direct ET | Adaptive evolution of *C. cellobioparum* | T = 10 g L$^{-1}$ | [87] |
| *Propionibacterium freudenreichii* | Glycerol & propionate; Only Propionate; Lactate & propionate | Acetate | Ferricyanide mediated ET | Enhanced bacterial growth & substrate consumption | Y = 56% $T = 0.38$ g L$^{-1}$; Y = 68% $T = 0.47$ g L$^{-1}$; Y = 60% $T = 0.42$ g L$^{-1}$ | [81] |
| *Enterobacter aerogens* NBRC 12010 | Glycerol | Ethanol; Hydrogen | Thionine mediated ET | Increased glycerol consumption | Y = 92% $T = 3.93$ g L$^{-1}$; Y = 74% $T = 0.14$ g L$^{-1}$ | [81] |
| *Cellulomonas uda* + *Geobacter sulfurreducens* | Cellobiose | Ethanol | Direct ET | Adaptive evolution and deleted *G. sulfurreducens* hydrogenase gene | No | [88] |

**Table 1.** *Cont.*

| Microbe | Substrate | Product | Mechanisms of EET | Genetic Modification of Host | Yield (Y) and/or Titre (T) | Ref. |
|---|---|---|---|---|---|---|
| **Anodic Electro-Fermentation** | | | | | | |
| *Ralstonia eutropha* | Fructose | Poly hydroxy butyrate | poly (2-methacryloyloxyethyl phosphorylcholine-co-vinyl-ferrocene)-mediated ET | Not modified | No | [89] |
| *Escherichia coli* | Lactate | Acetate; Ethanol | Direct ET | Introduction of *S. oneidensis* MR-1 *Mtr* pathway | Productivity = 0.038 mM day$^{-1}$; T = 40 $\pm$ 3 $\mu$M | [90] |
| *Escherichia coli* + *Methano bacterium formicicum* | Glycerol | Ethanol; Acetate | Methylene blue-mediated ET | Cyt c introduction—*CymA, MtrA* and *STC* from *S. oneidensis* | Y = 35% $\pm$ 5% T = 55.25 $\pm$ 7.76 g L$^{-1}$ Productivity = 12.12 $\pm$ 1.70 mg h$^{-1}$; Y = 20% $\pm$ 1% T = 40.75 $\pm$ 2.37 g L$^{-1}$ Productivity = 8.94 $\pm$ 0.52 mg h$^{-1}$ | [91] |
| *Pseudomonas putida F1* | Glucose | 2-Keto-gluconate | 7 different mediators-based mediated ET | Not modified | Y = 90% $\pm$ 2% T = 1.47 $\pm$ 0.27 g L$^{-1}$ Productivity = 1.75 $\pm$ 0.33 mg h$^{-1}$ | [92] |
| *Pseudomonas putida* | Glucose | 2-ketoglu conic acid | Direct ET | Overexpression of periplasmic glucose dehydrogenase | Productivity = 0.25 $\pm$ 0.02 mmol g$_{CDW}^{-1}$ h$^{-1}$ | [93] |
| *Corynebacterium glutamicum* + *Zymomonas mobilis* | Glucose Glucose | L-lysine; Ethanol | Ferricyanide-mediated ET Methyl naphthoquinone, humic acid, methylene blue, neutral red, 1,4-riboflavin, butane-disulfonate and tempol-mediated ET | Feedback deregulated mutant and overexpressed redox-related genes—*ZMO0899, ZMO1116* and *ZMO1885* | T = 2.9 Mm Productivity = 0.2 mmol L$^{-1}$ h$^{-1}$ Bioelectricity generation = 2.0 m Wm$^{-2}$; T = ~ 42.5 g L$^{-1}$ | [94] |
| **Microbial electrosynthesis (MES)** | | | | | | |
| *Clostridium pasteurianum DSM 525* | Glucose; Glycerol | Butanol; 1,3-propanediol | Direct ET | Not modified | T = 1.00 $\pm$ 0.20 g L$^{-1}$; T = 4.74 g L$^{-1}$ | [47] |
| *Geobacter sulfurreducens* | $CO_2$; Succinate | Glycerol | Direct ET | Not modified | T = 8.7 $\pm$ 0.3 mM | [95] |
| *Sporomusa ovate* + *Methanococcus maripaludis* | $CO_2$ | Acetate; $CH_4$ | $H_2$-mediated ET | Not modified | T = 0.2 to 0.3 mM; T = 0.2 to 0.3 mM; | [34] |
| *Sporomusa ovate* | $CO_2$ | Acetate | Direct ET | Not modified | No | [96] |

Table 1. *Cont.*

| Microbe | Substrate | Product | Mechanisms of EET | Genetic Modification of Host | Yield (Y) and/or Titre (T) | Ref. |
|---------|-----------|---------|-------------------|----------------------------|---------------------------|------|
| *Shewanella onedensis MR-1* | Acetoin | 2,3-butanediol | Direct ET | Heterologous expression of butanediol dehydrogenase (*Bdh*) gene along with a light-driven proton pump and hydrogenase gene $\Delta hyaB\Delta hydA$ knockout | T = 0.03 mM | [97] |
| **Anodic Electro-Fermentation** | | | | | | |
| *Clostridium pasteurianum* | Glycerol | 1,3-propanediol; n-butanol | Neutral red and brilliant blue-mediated ET | Not modified | Y = 0.41 mol mol$^{-1}$ glycerol in brilliant blue-mediated ET; Y = 0.35 mol mol$^{-1}$ glycerol in Neutral red-mediated ET | [98] |
| *Saccharomyces cerevisiae* | Dhea | 7α–OH–DHEA | Neutral red and 7α-hydroxylase-mediated ET | Heterogenous expression of 7α-hydroxylase | T = 288.6 ± 7.8 mg L$^{-1}$ | [99] |
| *Ralstonia eutropha* | CO$_2$ | Iso-propanol | H$_2$-mediated ET | Not modified | T = 216 mg L$^{-1}$ | [100] |
| | | 3-methyl-1-butanol; Isobutanol | Formate mediated ET | Introduction of genes *alsS, ilvC, ilvD, kivd* and *yqhD*; Knockout of polyhydroxy butyrate synthesis gene cluster (*phaC1, phaA* and *phaB1*) | Both depicted a titre of 140 mg L$^{-1}$ | [79] |
| *Xanthobacter autotrophicus* | N$_2$ and H$_2$O | NH$_3$ | H$_2$-mediated ET | Not modified | T = ~ 0.8 mM | [101] |

### 2.5. Increasing Electrode Interaction

The relationship between microbes and electrodes relies mainly on the cohesive nature of the biofilm, the electrode and how these species interact with the electrode [109]. To implement a variety of microbes, the electrodes have been improvised, yet, further research on strain modification for pure cultures is still necessary [110]. Two frequently used strains for understanding electrode interactions are *S. oneidensis* and *G. sulfurreducens*. In a related review, the latter bacterium was altered by deleting the genes that encode for a protein controlling the *Pilz* domain, forming a more coherent and conductive biofilm. Further, this mutant produced 6-fold more conductive biofilm when compared to its wild type. More production of pili led to a smaller potential loss. In another experiment, the former bacterium was altered to allow the production of biofilm with the help of heterologous overexpression of the cyclic di-GMP pathway gene that originated from *E. coli*. Hence, after a few hours, the collection of cells and electrolytes from the electrode depicted a significant change against the wild type of enhanced electrode [68,111–113].

### 3. Techniques for Improving MES Performance

It is imperative to enhance the MES's performance and optimise it while maintaining a low budget [114]. During the anodic and cathodic processes, the electron transfer system of bacteria is likely to follow their different path. So, the electrode materials don't need to yield good results in both the microbial cathode and microbial anode equally [115]. Various factors (physical, chemical and biological) affect MES differently, these have been depicted in Figure 4.

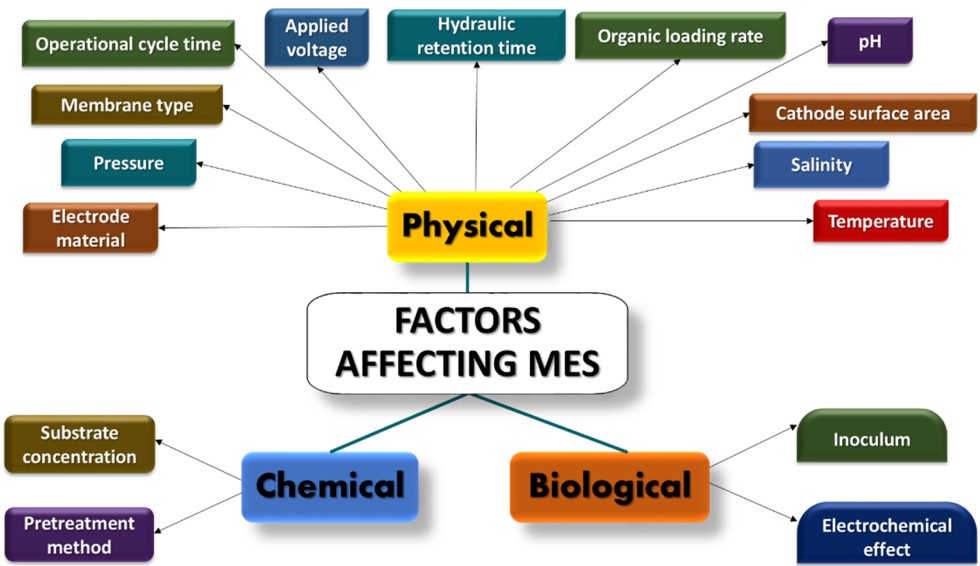

**Figure 4.** Several physical, chemical and biological factors affecting MES.

### 3.1. Cathode Fabrication

Since 2010, numerous commercially accessible carbon electrodes of various shapes and forms (like rods, fabric, block, AC (activated carbon) plates, gas diffused AC, reticulated vitreous carbon (RVC) and granules of fibre felt) have been developed for direct $CO_2$ reduction. Important features like chemical tolerance to degradation, cost-effectiveness, biocompatibility with long term and proven application of bioanodes make the cathodes more efficient and drive a better and broader usage in the long run in MES processes. For instance, industrially commercialised carbon material like graphite, carbon plate and carbon fabric and 3D structures such as carbon felt and carbon fibre rod electrodes are used in MES processes [2,48,61,116].

The key reason behind introducing surface modified materials is to promote cathodic biofilm for efficient $CO_2$ reduction in MES. The key to enhanced bioproduction lies in the

interaction between the bacteria and the cathode, therefore it is a requisite to choose the material and build a suitable cathode [117]. Carbon felt, mesh and cloth are commonly used to create surface-modified cathodes due to their various advantages, including high porosity, resilience, larger surface area and many more. Other features and characteristics of cathode and design modification to produce charge are complicated because bacterial attachment, electrode microbe rate of electron transfer, selective development of biofilm and maximum production rate cannot be sufficiently increased by treating the electrode with melamine and ammonia, against unmodified carbon cloth [118,119]. Compared to untreated graphite, the microwave treatment of nickel nanowire coated graphite increases the surface roughness by about 50 folds. Other modified electrodes include carbon cloth modified by utilising nanoparticles like gold, palladium, nickel or cotton and polyester-modified carbon nanotubes-coated cathodes (considered the most promising materials) or NanoWeb-RVC-cathodes that improve acetate output by 33.3 folds relative to unmodified carbon plates [48,118–121].

Biocathodes are electrodes enriched with microorganisms, with whose help they perform reductive reactions for various substrates. These electrodes use the metabolic activity of the microbes for the same purpose. Cai et al. [122], in a bioelectrochemical method, implemented a biocathode to increase the oxygen reduction and to generate electricity in air cathode MFC. They used it with water for the removal of contaminants present in the catholyte solution and used it for the production of target commodities such as $H_2$ production using protons as $e^-$ acceptors. Kondaveeti and their team [123] utilised biocathode in BES to reduce $CO_2$ to form products (methane, VFAs and alcohols). Tahir and his team [124] used MXene–coated biochar as an MES biocathode for selective VFA production. Improvement of the biofilm development, attachment of bacterial cell, rate of $e^-$ transfer at cathode surface, as well as the rate of chemical production, requires key elements like best cathode materials, selective microbial groups and well-organised reactor design. There are various reports and studies on new materials of electrode discovery and modification systems of surface for anodic process development (See Table 2) [115].

**Table 2.** It depicts the carbon cloth cathode treatment and acetate production rate for each day including consumption density [16].

| Carbon Cloth Cathode Treatment | Average Current Consumption Density (mA m$^{-2}$) | Acetate (mM m$^{-2}$ day$^{-1}$) | Coulombic Efficiency |
|---|---|---|---|
| Carbon cloth | −71 ± 11 | 30 ± 7 | 76 ± 14 |
| 3-Aminopropyltriethoxysilane | −206 ± 11 | 95 ± 20 | 82 ± 11 |
| Ni | −302 ± 48 | 136 ± 33 | 80 ± 15 |
| Melamine | −69 ± 9 | 31 ± 08 | 80 ± 15 |
| Carbon Nanotube-cotton | −220 ± 1 | 102 ± 25 | 83 ± 10 |
| Cyanuric chloride | −451 ± 79 | 205 ± 50 | 81 ± 16 |
| Ammonia | −60 ± 21 | 28 ± 14 | 82 ± 8 |
| Pd | −320 ± 64 | 141 ± 35 | 79 ± 16 |
| Chitosan | −475 ± 18 | 229 ± 56 | 86 ± 12 |
| Polyaniline | −189 ± 18 | 90 ± 22 | 85 ± 7 |
| Au | −388 ± 43 | 181 ± 44 | 83 ± 14 |
| Carbon Nanotube-polyester | −210 ± 13 | 96 ± 24 | 82 ± 8 |

### 3.2. Anode Fabrication

According to various studies, carbonaceous materials like carbon cloth, glassy carbon, graphite felt, carbon, granules of graphite and rods are mainly used as the anode material

and are available commercially. These materials have advantages in chemical stability and good conductivity of electricity. There are various methods available to enhance the formation of biofilm and MFC performance. The primary is to increase the attachment of bacteria with a bacterial electrode via EET. In this electron transfer, the potential is kept at enzyme redox potentials rather than the addition of mediator potential and exogenous mediators. For example, the node is pre-treated with ammonia gas, aqua fortis and ethylenediamine at 700 °C, with $HNO_3$ and quinone or quinoids, showing a spike in anode density of the microbial cell at a successful rate. $HNO_3$ and hydrazine can also be used as alternatives [125,126].

Second is the electrode pretreatment before displaying them to MFC operation and biofilm development. Other pretreatment methods are also available that aim at changing the surface of the electrode with different redox molecules, this helps in the transfer of an $e^-$ from the microorganism to the electrode. Biofilm formation is also a difficult process and affects different factors like charge, hydrophobicity, topography along with bacterial properties and environmental factors [127,128]. Several studies suggest that the positively charged surface of anode allows for higher attachment of bacteria and biofilm activity. When charged bacterial surfaces are suspended in the aqueous suspension, then the charged surface attracts more bacterial cells. The positive surface charge on the carbon clothes was extended from 0.38 to 3.99 meq m$^{-2}$ when treated with ammonia which successfully reduced the acclimation time of microbial by 50% and maximal power density increased. Bacteria are more attracted to groove or braided surface rather than a smooth surface to colonise porously and adhere easily to the surface as rough surface is the more favourable site for colonisation [16,129–132]. MFC performance can be enhanced by expanding the available expanse for biofilm growth using rough or porous materials [128].

### 3.3. MES and Gene Manipulation

A gene manipulation technique via systems biology approach must be carefully designed and tested to determine which gene of the microbe should be altered and for what purpose to enhance MES. The strategy which can be adopted is the microbe's EET efficiency improvement to increase the cathodic chamber activity. Alternatively, a different pathway can be used for yielding valuable products by choosing the heterologous or homologous expression. The appropriate method can be dependent on whether or not the target strain is simple to adjust (based on the tools available). One of the most representative MES capable acetogen is *Sporomusa ovata*, but in a recent study, *C. ljungdahlii* were applied [24,133,134].

### 3.3.1. Modification of Pathways for Generating Value-Added Products

When the microbe capable of binding to the electrode is incompetent to create enough of the desired output but can use another pathway to produce the desired result, modification of the pathway for the processing of value-added commodities becomes essential. The alternate method could be exogenous, producing a heterologous effect or competitiveness, suppressing its gene system. Microbe such as *Clostridium ljungdahlii* has often been used due to the presence of a well-established tool that allows deletion of the gene [135].

Genetically modified (GM) microorganisms mentioned in this subsection have not been introduced in MES systems yet. Nevertheless, these still have enough potential of being MES-capable microbes because they contain most of the $e^-$ transfer systems proposed in *C. ljungdahlii*. Through heterologous expression, this strain was altered for generating butanol from the initial *Clostridium acetobutylicum*. The resulting strain increased the production of butanol significantly. Nevertheless, *C. ljungdahlii* transformed butanol to butyrate and so no butanol was found at the end of the process. In the experiment, the presence of butanol was confirmed due to the active expression of microbe genes [136–138]. In another research, a lactose inducible mechanism designed for *Clostridium perfringens* was incorporated into *C. ljungdahlii*. This arrangement improved the production of ethanol by 30-fold [139]. Apart from advantages like easy optimisation for highest yields, high selectivity, providing resistance against system fluctuations and $O_2$ intrusion and facili-

tating a wider product spectrum of high-value molecules; GM microbes also have a few limitations. The most prominent limitation being questionable societal acceptance and grant of approval by the government [8].

### 3.3.2. Host Cell Selection

Concerning adaptation to other areas or even new trials, GM MES is only in its early stages, holding an infinite scope of research. Genetic modification is a method used in biotechnology to recreate host cell DNA and perform insertion and deletion of genes or producing point mutation through homologous or heterologous expression. Therefore, when analysing the possibility of GM in MES, the preference of the host cell is a primary concern. The selected host organism that is to be subjected to genetic modification should have a simple and fully sequenced genome along with the required genetic tool. Additionally, microorganisms that have more advantages and all the other necessary features are favoured for MES. Three principal aspirants from our perspective for MES are *E. coli*, *C. ljungdahlii* and *Cyanobacteria* [18,62,63,140]

*Escherichia coli*

One of the most commonly used laboratory microbes in the field of biotechnology is *E. coli*. GM *E. coli* is a crucial microorganism that plays a central role in the generation of heterologous proteins, like the development of vaccines, bioremediation and much more [141,142]. Genetic tools present in this organism can be exploited in the field of MES. Neutral red is used as an electron shuttle in *E. coli*, aiding in generating more products such as pyruvate, lactic acid and succinic acid when compared to usual conditions [62,63,91,143]. Further, this bacterium is commonly utilised for heterologous expression to find out the features of redox protein that take part in the process of e$^-$ transfer [144]. Hence, suggesting that *E. coli* has ample capabilities to function as a host cell in the MES field.

*Clostridium ljungdahlii*

*C. ljungdahlii* holds high worth as a host cell not only because it is a potential MES-capable microbe, but also due to its ability to reduce $CO_2$, treat waste gas and produce various products, like ethanol and acetic acid via fermentation [135,145]. Hence, *C. ljungdahlii*-driven MES can be used as a cell factory with continuous treatment of industrial waste gas. Modifying its gene proves that it is capable of generating diverse multi-carbon products as described earlier [139].

*Cyanobacteria* (Cyano)

The only photosynthetic prokaryote capable of extracting oxygen by splitting water is cyanobacteria, which shares several benefits with microalgae [146]. Additionally, this blue-green algae are a crucial third-generation biomass producer because of its capability to photosynthesise (oxygenic), non-food-based feedstock, high per-acre productivity and land-independent growth [147,148]. Research based on intracellular and extracellular e$^-$ transfer in Cyano has revealed that they vastly differ from other microbes used in MES [149]. Hence, as a host cell for MES, these next-generation biomass producers hold drastic value in the field of research. Appropriate and enough data is needed to develop an efficient strategy. Nevertheless, it is almost impossible to obtain nearly all the in vitro and in vivo data due to a lack of room and adequate time. A recent development in the bioinformatics field offers the perfect method known as "in-silico" to solve issues related to the collection of data. Numerous aspects and features of the microbe could be analysed using the in-silico method. These research-based in-silico methods present the forecast results on ATP yield, biomass, $CO_2$ fixing pathway, reduction degree of the product and substrate and fluctuations in the flux via specific pathways [150]. Therefore, evaluation based on in-silico methods can help in providing vital data to develop an efficient strategy of metabolism.

## 4. MES Allows Biocatalysts to Utilise $CO_2$ and Generate Electricity

These days, the most important global problem is the elevated $CO_2$ emissions that cause a spike in the average global temperature. Greenhouse gases play a central role in emergent global warming. There are many greenhouse gases responsible, but among those, the sole contribution of $CO_2$ is 63%, which is quite high. This has led to the emergence of the use of different $CO_2$ capturing and storing techniques. Various $CO_2$ utilisation procedures have been discovered that have been used for hoarding and transforming $CO_2$ into high-valuable products via MES by applying bio-electrochemical techniques using electricity as the source of energy [8–10,151–154].

MES focuses on using renewable sources rather than using non-renewable sources such as crude oil, decreasing the usage of naphtha-based chemicals. MES is not harmful to the environment, making it useful for future production of $CO_2$ and the protection of the environment by $CO_2$ sequestration, averting various environmental issues. We are aware of the thermodynamic stability of $CO_2$, it requires a supply of external energy for the activation and various conversion reactions. MES set-up usually comprises a couple of chambers called abiotic anodic chamber and biotic cathodic chamber that contains proton exchange membrane (pEM) aiding in the movement of protons across the two chambers (See Figures 2 and 3). In the anodic chamber, water molecules split into protons ($p^+$), $e^-$ and gaseous $O_2$ is released. The $p^+$ are transferred to the cathodic chamber via the pEM and the $e^-$ are drawn to the cathode through an external circuit. In the cathodic chamber, the $e^-$ and $p^+$ or energy carriers such as $H_2$ and $CO_2$ are integrated by biocatalysts to produce volatile fatty acids (VFAs), alcohols, butyrate, formate, acetate, etc. via $H_2$ mediated $e^-$ transfer or DET. Generally, the acetogens employ the Wood–Ljungdahl (WL) pathway for $CO_2$ fixation and are used as biocatalysts at the cathode, yielding acetate as the chief MES product, nevertheless, other organic chemicals with more carbon content, like butyrate, caproate, ethanol, caprylate, propionate and isopropanol can also be produced [25,114,155].

For breaking the double bonds of $CO_2$ a massive quantity of energy is needed as $CO_2$ is thermodynamically very stable. Therefore, metal catalysts can be implemented for reducing the energy to be employed for cleaving the bonds. This bottleneck can be overcome by using different and better catalysts that are easy to manipulate or biocatalysts in the form of enzymes or microbes, reducing the economic feasibility of the process. Microorganisms responsible for reducing $CO_2$ into organic compounds by consuming $e^-$ are termed electroautotrophs. These types of microorganisms can survive as biofilms and also as planktonic cells in the bulk phase. Using biocatalysts for $CO_2$ reduction can be beneficial in operating the process and capital costs in the process. MES also has a few drawbacks, it currently has low product yield, the product once obtained needs to undergo downstream processing (purification and separation); its high capital cost makes scaling up challenging; and lastly, longer carbon chain chemicals have a low production rate [2,8,19]. A comparative synopsis of several microbial catalysts used, cathode materials employed and yield of product obtained via MES has been depicted by Jourdin and his team [19].

## 5. Diverse Products Obtained from $CO_2$

Technologies like the MES are the need of the hour, as they not only focus on the high efficiency of the system but also aid in the production of a variety of products by reducing and converting $CO_2$ in MES [2,8,10,15,17,156]. One of the most common by-products is acetate [157]. Alternatively, butyrate, oxobutyrate, ethanol and isopropanol can also be produced using MES [20,158]. It was also observed that on further reduction of acetate that has been generated and accumulated in the system, more commodities can be produced [157]. In the case of ethanol, it was observed that organisms such as *S. ovata*, when kept under highly reductive conditions in the presence of excess $O_2$, generated ethanol [22,159]. Often a variety of alcohols and VFAs have been observed during MES production [160,161].

Acetyl-CoA is the primary precursor in $CO_2$ reduction that takes place via the Wood-Ljungdahl (WL) pathway [162]. This precursor molecule then undergoes several steps

and metabolic changes to produce diverse chemicals [163]. Acetogenic microbes are used for this process to reduce $CO_2$, using $H_2$ as the $e^-$ donor. By modifying the conditions under which the system is operated, mechanisms such as solventogenesis metabolism can also be introduced in MES [164]. An important microorganism used for this process is *Clostridium ljungdahlii*. Although pure cultures can be used for this process, even mixed cultures are an amazing alternative, as portrayed in Table 3 [25,145,165,166]. Apart from ethanol, butyrate and acetate, lactate and succinate can also be produced from intermediate products generated in the system via the Krebs cycle. To summarise, MES produced precursor compounds such as acetate, which can further be upgraded to longer chain fatty acid, biofuels, bioplastics and so on, via multi-step conversions. For example, acetate on being upgraded can produce butyrate and caproate via the chain elongation method in MES. Some other examples include the production of methanol, formaldehyde and ethylene from methane and the production of single-cell proteins and polyhydroxyalkanoates from short-chain fatty acids [20,114,158,163,167].

**Table 3.** Microorganisms were tested by Nevin and his team [96] to check the solid-state electrode's ability to receive electrons and to use $CO_2$ as the terminal electron acceptor, following the Wood-Ljungdahl pathway.

| Species | Electron Consumption? (EC) | EC Rate vs. *S. ovata* | Electron Recovery in Products | Products Formed |
|---|---|---|---|---|
| *Moorella thermoacetica* | Yes | - | 85% ± 7% (n = 3) | Acetate |
| *Clostridium lijungdahli* | Yes | - | 82% ± 10% (n = 3) | Acetate + Minor formate and 2-oxobutyrate over time |
| *Sporomusa ovata* | Yes | 100% | 86% ± 21% | Acetate + Trace of 2-oxobutyrate |
| *Acetobacterium woodii* | No | - | - | - |
| *Sporomusa silvacetica* | Yes | 10% | 48% ± 6% | Acetate + Trace of 2-oxobutyrate + non-identified products |
| *Clostridium aceticum* | Yes | - | 53% ± 4% (n = 2) | 2-oxobutyrate and acetate as prime products and other non-identified |
| *Sporomusa sphaeroides* | Yes | 5% | 84% ± 26% (n = 3) | Acetate |

Nevin and his team [96] were the first to prove that biocathode systems can be used for the reduction of $CO_2$ and to yield acetate. Some microorganisms are known to obtain the carbon for metabolic processes from the atmospheric $CO_2$ and some from inorganic sources. Both pure and mixed cultures can be used in this process, but mixed cultures are more preferred as they can be obtained in large quantities simultaneously and can easily tolerate environmental conditions as compared to pure cultures which need a specific growth medium and are vulnerable to system fluctuations and $O_2$ intrusion [2,8]. Till now, both mixed and pure cultures have shown similar recovery of $e^-$. $CO_2$ can be reduced to acetate with the aid of acetogenic bacteria that uses hydrogen as an electron donor. The pure culture of this bacteria is poured into the cathodic compartment that has been already filled with a mixture of gas and various $e^-$ donors for increased and sufficient growth of the culture on the electrodes. $H_2$ production was controlled by applying $-400$ mV potential (vs. SHE) to the cathode, meanwhile, after switching the gas feed to $N_2$-$CO_2$, the acetate was produced along with the small volume of 2-oxobutyrate that had $e^-$ recovery up to 85%. The cathode biofilms used were long-lasting as they were capable of accepting the $e^-$ after 3 months and could also produce acetate but, the acetogenic microorganisms lack this property and gained very little energy by the reduction of $CO_2$, which implies the use of substantial energy inputs or expensive catalysts which seems impractical. The researchers further tried experimenting with other microbial species to check whether they were capable of MES or not, several other acetogenic bacteria were able to gain electrons

at the electrode. As seen in Table 3. the acetogen *Sporomosa ovata*, a close relative to *M. thermoacetica* was able to directly accept e$^-$ from the cathode and transform $CO_2$ to acetate and 2-oxobutyrate, whereas *A. woodii*, was unable to do so as it lacked Cyt c and relies upon the sodium gradient which is coupled to the WL pathway, thereby, reflecting a different behaviour as compared to other acetogens in Nevins experiment [24,25,78].

### 5.1. $H_2$ Production via MES

Hydrogen is a valuable fuel that can be produced efficiently by MEC. Materials used as cathode catalysts include platinum for the production of $H_2$ from MES, but because of its high cost, it is not viable economically, so other alternative materials like stainless steel and nickel are used as these have low cost, stability and low over-potentials. For $H_2$ production, nickel and stainless steel have more efficiency against platinum owing to their low voltage and cheap cost [32,34,168–170].

But enzymatic biocathodes are unstable and not self-generating, hence they lose their activity of catalysis over time. The study suggested that $H_2$ production is successfully catalysed by immobilising the enzymes responsible for catalysing the reversible reaction at carbon electrodes. *Desulfovibrio* species (hydrogenases processing microbe) are used for hydrogen production by immobilising methyl viologen that acts as a redox mediator. Mixed cultures can be employed to enhance microbial $H_2$ production as they show more desirable characteristics like steadiness and relevancy in BES. Acetate and hydrogen are used as an e$^-$ donor that changes the electrode polarity and with anode attached biochemically-active biofilm reverses the biocathode's mode for $H_2$ production [171–173]. Examples of studies on $H_2$ production using several substrates and VFA mixtures in MECs have been reviewed by Rivera et al. [37] and Cardeña and team [36].

### 5.2. Acetate Production via MES

For acetic acid, the production of the NanoWeb-RVC (carbon nanotubes on reticulated vitreous carbon) showed very high efficiency as a biocathode component. This type of electrode is advantageous for macro structured RVC and nanostructured surface modification. Effective mass transfer is ensured to and from the biocatalyst due to the high surface area to volume ratio of the macroporous RVC. The carbon nanostructure increases microbial EET, improves the interaction between microbe and electrode, enhances the development of microbes and helps in bacterial attachment. So NanoWeb-RVC displays a high intrinsic performance as a biocathode component for MES and is considered an effective material from an engineering perspective.

Electrophoretic deposition (EPD) is a method in which colloidal solution is utilised to make thin films. It has also been used on a large scale to make highly porous electrodes for electrochemical applications from the deposition of carbon nanotubes (CNT). For processing the CNTs, EPD is the easiest process to operate that employs simple equipment. However, it can also produce narrow films from colloidal suspensions on substrates irregular in shape. An increase in production can be achieved just by expanding the dimensions of the existing substrate to be coated. So EPD demands bulky and industrial-scale manufacturing of porous electrodes. The MES has been recorded to achieve a high acetic acid production rate of up to $685 \, \mathrm{g \, m^{-2} \, day^{-1}}$ from $CO_2$, using enriched microbial culture and a newly synthesised material for the electrode [16,61,174,175].

The study by Tian et al. [168] demonstrated that the MES performance can improve by hydrogen evolution reaction catalyst (HER). This involved the construction of a molybdenum carbide ($Mo_2C$) modified electrode, an active HER electrocatalyst, the final acetate yield rate of MES is much higher. Electrochemical studies and analysis also suggested that $Mo_2C$ can be induced for the production of $H_2$ and help in the biofilm formation and monitor the mixed culture of microbes. It shows an electronic structure similar to the metal group like platinum, considering the high performing HER electrocatalyst. The presence of molybdenum carbide in carbon felt ($Mo_2C$-CF) results in increased evolution of $H_2$ in the MES, which averagely shows 12.7 times higher than CF without $Mo_2C$. The presence of

$Mo_2C$ also helps to regulate the mixed culture of microbes in biofilms and the planktonic cells in microbial electrosynthesis. Some of the microbes involved in the MES system are namely *Acetobacterium*, *Citrobacter*, *Arcobacter*, etc. $H_2$ acts as the $e^-$ carrier and helps in $e^-$ transport through a hydrogen-related metabolic system due to the presence of HER electrocatalyst cathode in the MES. This also helps in the $CO_2$ reduction step in MES due to the coupling of an active HER cathode. Refer to Table 4 to compare the yield of acetate when a mixed microbial flora is employed for MES.

The coupling of molybdenum carbide in CF cathode is one of the most vital, rapid and simple studies that efficiently improve the MES system. To develop a highly efficient $H_2$ catalyst, a neutral condition but the HER electrocatalyst of $Mo_2C$ reported the advantages of hydrogen evolution even in the acidic condition. Therefore, the presence of active HER catalysts like $Mo_2C$ increases the release of hydrogen, which helps the growth of the biofilm of mixed microbial culture, and thus resulted in a higher reduction rate of $CO_2$ and generation of acetate in the MES system [168,176].

**Table 4.** Review of literature on the yield of acetate via mixed microbial flora in MES.

| Cathode Material | $E_{cathode}$ (V vs. SHE) | Current Density (A m$^{-2}$) | Volumetric Production Rate (g L$^{-1}$ day$^{-1}$) | Maximum Acetate Titre (g L$^{-1}$) | Coulombic Efficiency (%) | Ref. |
|---|---|---|---|---|---|---|
| 12 mg cm$^{-2}$ $Mo_2C$ | −0.85 | −5.2 | 0.19 | 5.72 | 64 | [168] |
| NanoWeb-RVC | −0.85 | −37 | 0.03 | 1.65 | 70 | [61] |
| Graphene-nickel foam | −0.85 | −10.2 | 0.19 | 5.46 | 70 | [177] |
| VITO-CoRE™ electrode fabricated with activated carbon | −0.6 | −0.069 | 0.14 | 4.97 | 45.5 | [178] |
| Carbon felt (CF) | −1.26 | −5.0 | 0.06 | 1.29 | 58 | [179] |
| | −0.903 | −2.96 | −0.14 | 4.7 | 89.5 | [180] |
| CF and stainless steel | −0.78 | −15 | 0.14 | 2 | 22.5 | [7] |
| | −0.9 | −10 | 1.3 | 0.6 | 40 | [181] |
| RVC-EPD | −0.85 | −102 | - | 11 | 100 | [182] |
| rGO-CF | −0.85 | −4.9 | 0.17 | 7.1 | 77 | [177] |
| CF with fluidised GAC (16 g L$^{-1}$) | −0.85 | −4.08 | 0.14 | 3.9 | 65 | [183] |
| Graphite stick-graphite felt | −0.8 | −20 | 0.14 | 8.28 | - | [184] |
| Graphite granules | −0.6 | - | 1.0 | 10.5 | 69 | [185] |

*5.3. Formic Acid Production via MES*

$CO_2$ can be converted into liquid formic acid with the help of sustainable electricity, which later can serve as a chemical for preserving food, alternative future fuel and an energy storage molecule. Formic acid has been derived primarily from fossil reserves, which are estimated to get depleted, to solve this problem green alternative ways have been discovered through microbial transformations. Formic acid production was earlier carried at laboratory scale using $CO_2$ and electricity, a direct electrochemical conversion where $H_2O$ is dissociated into $H_2$ and $O_2$, and the former is then used to reduce $CO_2$ into formic acid. Production of formic acid requires less energy as compared to methane and methanol production against $CO_2$ [154,186,187].

At a commercial scale, formic acid can be produced from methanol via multiple pathways, initially, it is transformed to methyl formate followed by hydrolysis to produce formate. Later on, formate production via direct conversion is analysed through hydrogenation; nevertheless, the end product is formate. Therefore, a single-step chemical reaction has been recently discovered, where formate is produced via electrochemical reduction of $CO_2$ by $H_2$ generated from $H_2O$. The electrochemical set-up consists of an anode and cathode, where the hydrolysis of water and formation of formate takes place. Different electrode systems have been used for direct electrochemical reduction of $CO_2$, including metals, nonmetals and bioelectrodes. The electroreduction potential of −1.85 V vs. SHE is required for yielding formic acid. However, different types of compounds can

be formed in MES, including various hydrocarbons like alcohols and carboxylic acids such as butyric acid. Various studies have revealed the selective production of formic acid from $CO_2$ using different electrode materials, but the products generated are mostly non-specific. The properties of formate of high solubility, easy conversion into other compounds and its decomposition at the anode are responsible for the lower yield and increase in separation cost. These limitations have been resolved by utilising enzymatic $CO_2$ electroreduction for the generation of formic acid. The key enzyme involved is formate dehydrogenase, it catalyses the oxidation of formate to $CO_2$, and reduces $CO_2$ to formate, which is chaperoned by the $NAD^+$ to NADH redox cycle [188–190].

*5.4. Syngas Production via MES*

To satisfy the potential demand for biomethane and energy, the present supply of organic waste is not sufficient, hence it is necessary to increase the output to fulfil this demand [191,192]. The organic sources of methane are limited, although enough $CO_2$ is present from the industrial exhaust, electrons can also be acquired from water, sulphides and ammonium. Further, methane can directly be converted to syngas through hydrogenotrophic methanogenesis or it can also be produced indirectly by using acetate as an intermediate [192,193]. Syngas or synthetic gas is primarily a combination of gases such as $H_2$, CO and sometimes $CO_2$ and is used for electricity generation [191,194]. The anodic and cathodic chambers were constructed keeping in mind the reactor volume. This would provide an optimum surface area for the system, hence making the process more efficient [192].

Initially, when syngas fermentation was combined with a single cell anaerobic digestor (AD-MES system) [192], an experiment of three phases was conducted where the anodic and cathodic chambers were set up considering the volume of the reactor. In phase I, the experiment was conducted inside a glass reactor which was a lab-scale fermentation reactor. The II phase triggered the open circuit in which electrodes were established in the glass reactor through phase 1 and fresh inoculum was added. In phase III voltage of −0.8V vs. SHE was applied to the syngas from phase II for the production of gases like methane, carbon dioxide and hydrogen [195–197]. The conclusion of this experiment built the starting point of combining two processes i.e., syngas fermentation with a single cell AD-MES system [195]. Examples of production of syngas (such as biomethane and biohydrogen) and value-added biochemicals (such as $H_2O_2$, bioalcohols, acetate and VFAs) using BESs have been briefly summarised by Kumar et al. [198].

Coupling Anaerobic digestion (AD) with MES is one of the most novel technologies through which $CO_2$ can be generated and can further be reduced to methane with the help of microbes [195]. Similarly, using the same system, the $CO_2$ present in the biogas can be converted to acetic acid or other chemicals with the help of chemolithoautotrophic microorganisms. The main benefit of using this combination (AD-MES) is that the system is cost-effective and requires low capital, simultaneously the system also continuously produces and upgrades biogas while using only small energy input. The biogas generated by anaerobic digestion contains approximately 40–60% of $CO_2$, which can be utilised as a feedstock in the MES to generate diverse chemicals by reducing $CO_2$. Integrating both these processes has been proven to enhance the production of methane, hence enriching the composition of methane in biogas, as well as producing other value-added chemical commodities. Thus, offering additional economic benefits [33,48,51,199–201].

## 6. MES Enhancement

The $CO_2$ utilisation and the production of unsaturated VFAs on electrophoretic deposition and 3D-reactors, were continuously produced till the termination end of the trial, but the sole chemical compound generated was acetate, however, there was no accumulation of any other compounds like alcohols or VFAs. In the beginning, when the culture was transferred to the reactor, a steady production of acetate was observed concerning $CO_2$ consumption. The max average $CO_2$ utilisation rate observed was about 24.8 mol $m^{-2}$ $day^{-1}$

and the rate of acetic acid production was around 11.6 mol m$^{-2}$ day$^{-1}$ was reached, from 7 weeks onwards on EPD-3D [182]. Given a carbon balance, 94 $\pm$ 2% of $CO_2$ was discovered to be changed over to acetic acid derivation (with the rest of the carbon probably being utilised for biomass creation), while an e$^-$ balance uncovered a much all the more hitting result with 100 $\pm$ 4% of the electrons expended being recouped as acetic acid derivation. The changed regulations and item virtue accomplished in these investigations are outstandingly high, particularly for blended societies, which makes it intriguing for possible huge scope creation applications and downstream handling. Moreover, the accomplished acetic acid derivation production rate was around 685 $\pm$ 30 g m$^{-2}$ day$^{-1}$ is about 3.6 fold higher than the most noteworthy production rate [169] (Refer Table 4). Besides, a genuinely high acetic acid production titre of up to 11 g L$^{-1}$ was obtained, without any indications of item restraint of the dynamic microbes by then. It is along these lines very possible that the titre would have extended much higher qualities had the experiment not been halted. A high titre is a basic trademark for forthcoming enormous scope usage as it delivers the downstream processing a lot simpler than when the item fixation was low [169,182].

MES performance can be evaluated utilising a few key boundaries; these are recorded in Table 5, for most MES to acetic acid derivation is reported to date. The results summed up in the table are gathered from mixed as well as pure cultures of microorganisms in fed-batch or continuous mode. There are various types of materials of cathode and cathode applied. The potentials make it difficult to differentiate between the studies. For modern industries, the bioproduction process, for instance, in the fermentation production rate of 2 to 4 g L$^{-1}$ h$^{-1}$ having 99% yield is necessary for process feasibility. In the past decade, researchers have tried to access the scale-up viability of BESs specifically. However, considering the 3-dimensional nature of the electrode and its total surface area to volume unit of 2620 m$^2$ m$^{-3}$, attained 72% [19,169,177,182].

**Table 5.** Major performance factors of most MES to acetate researches reported until now. Adapted from [16,19,177,202].

| Microbial Inoculum | Cathode Material | $E_{cathode}$ (V vs. SHE) | Current Density (A m$^{-2}$) | Acetate Production (g$^{-2}$ day$^{-1}$) | Max Acetate (g L$^{-2}$) | Electron Recovery into Acetate % |
|---|---|---|---|---|---|---|
| *S. ovata* (continuous) | Graphite rods | $-0.4$ | $-0.208$ | 1.3 | 0.063 | 86 $\pm$ 21 |
| *C. ljungdahlii* (continuous) | Graphite rods | $-0.4$ | $-0.029$ | 0.14 | - | 72 |
| *Brewery WW sludge (fed-batch)* | Graphite granules | $-0.590$ | - | - | 1.71 | 67 |
| *Enriched Brewery WW sludge (fed-batch)* | Graphite granules | $-0.590$ | - | - | 10.5 | 69 |
| *Enriched WWTP sludge (fed-batch)* | Carbon felt | $-0.9$ | - | 34.5 | - | 89.5 |
| *S. ovata* (continuous) | Carbon cloth chitosan | $-0.4$ | $-0.475$ | 2.7 | 0.118 | 86 $\pm$ 12 |
| *S. ovata* (continuous) | CNT cotton CNT polyester | $-0.4$ | $-0.215$ | ~1.2 | 0.059 | 83 $\pm$ 10 |
| *S. ovate* (continuous) | Network coated graphite | $-0.4$ | ~$-0.625$ | 3.3 | - | 82 $\pm$ 14 |
| *Enriched Brewery WW sludge (fed-batch)* | Graphite rods | $-0.6$ | $-0.92 \pm 0.12$ | 8.56 $\pm$ 3.22 | - | 40 |
| *Mesophilic Brewery WW anaerobic sludge (fed-batch)* | Graphite felt | $-1.1$ | ~$-2.8$ | 10.1 | 1.4 | 65 |

**Table 5.** *Cont.*

| Microbial Inoculum | Cathode Material | $E_{cathode}$ (V vs. SHE) | Current Density (A m$^{-2}$) | Acetate Production (g$^{-2}$ day$^{-1}$) | Max Acetate (g L$^{-2}$) | Electron Recovery into Acetate % |
|---|---|---|---|---|---|---|
| *Anaerobic digester* | Graphite granules | −0.6 | - | - | - | 28.9 ± 6.1 |
| *Mixed natural & engineered sludge (fed-batch)* | NanoWeb-RVC | −0.85 | −37 | 192 | 1.65 | 70 ± 11 |
| *Enriched Mixed natural & engineered sludge (fed-batch)* | EPD-3D | −0.85 | −102 | 685 | 11 | 100 ± 4 |

## 7. Downstream Processes Involved in MES

Downstream processing of aimed complexes includes microbe, media and other by-products from the catholyte (electrolyte in the cathodic chamber). Hence, scholars have come up with new technology for distinguishing products, usually acetate extraction from the whole catholyte. Anion exchange resin (AER) is employed in extracting acetate in MES from the catholyte, the ratio of acetate in the solution was 16:4. AER can absorb 10 to 20 mg g$^{-1}$ acetate in just a single day from a catholyte broth comprising several kinds of compounds. Another technique is employed for the separation of butyrate from catholyte by using a membrane with a hollow fibre made up of propylene. Acetate can be separated by an alternative extraction method using an extraction chamber placed between an anodic and a cathodic chamber. Anion exchange membrane and PEM distinguishes the extraction chamber from the prior mentioned ones, only allowing passage of carboxylic acid to get deposited inside the extraction chamber. The concomitant production and separation of carboxylic acid have many privileges over the high capital and operational costs of using more than one set-up, providing economic sustainability to MES [19,31,161,203–206].

### 7.1. Process for Conventional Separation

The most commonly used process for separating organic acids is adsorption, here the ions are exchanged between carboxylate groups and functionalised solid sorbents. The activity of sorbents relies on the pH of the solution; when the pH is in intermediate capacity (~6.5), the adsorption increases, however at increasing pH the concentration of ionised acid will rise along with the decline in protonated amine concentration. Another excellent alternative technique employed in the conventional separation of organic acids is liquid-liquid separation. Several extracts like aliphatic amines and tri-n-octyl-phosphine are preferred for this process [207,208].

### 7.2. Pressure and Concentration-Driven Separation Process

In another extraction process (protraction) involving the immobilisation of organic solvents with the help of capillary action into the small pores of the hydrophobic micro-filtration membrane, the feed gets separated from the permeate. Diffusion of organic compounds occurs rapidly via organic solvent onto the membrane which can be extracted on the permeate side. This membrane only provides mechanical support, the extracts perform the chief function. The process of protraction is generally employed for extracting VFAs. It is the favoured process over liquid-liquid extraction as it is a simultaneous process, solvent stripping occurs, it is not expensive and the amount of solvent required is also little [209].

Higher mass transfer rates are achieved by changing the configuration of the membrane set-up oppositely while using hollow fibre membranes, where the shell side is for feeding of organic phase while the tube side is for the aqueous phase. Nowadays, silicon membranes are widely used for the extraction processes in which the water is utilised as an

extract and has portrayed excellent selectivity towards VFAs based on hydrophobicity. The process is overcome at a low pH as only un-dissociated acids are extracted. The selective extraction of butyric acid was achieved over acetic acid and propionic acid. However, this process is also reliable for alcohol extraction as the nutrients are preserved, making them feasible for extracting products from MES catholyte [155,210,211].

In the process of pervaporation, the process of partial evaporation is used to separate the compounds where the permeate side is kept under the influence of a vacuum. The extracts most commonly used in this process are the high-molecular-weight alkyl amines like tri-n-octyl phosphine oxide, trioctylamine and trilaurylamine [8]. The process of nanofiltration (NF) has been studied extensively for the extraction of VFAs. In NF, the effect of pH has been observed on the membrane charges as well as the degree of ionisation of acid. An increase in rejection of acetic acid has been seen (from 0–65%) when pH increases (3 to 7). The NF membrane is negatively charged, thereby, when the pH is increased the membrane restricts the carboxylate ions because of ongoing electrostatic effects. This makes a low pH for NF more beneficial for separating acetic acid [210,211].

### 7.3. Process of Reactive Extraction

To extract the organic acids, ionic liquids (ILs) have been studied. ILs were used to coincidently concentrate and esterify the acetic acid that was previously extracted via membrane electrolysis by Anderson and his fellow researchers [212]. He used bis (trifluoromethylsulfonyl)-imide IL, for concentrating acetic acid up to 80 Mm, when ethanol was added (max conversion of 90%), it was esterified to ethyl acetate. There is one other technique for the separation of VFAs, it is the usage of organic solvents in the presence of supercritical $CO_2$. For the extraction of propionic acid, trioctylamine was used in supercritical $CO_2$, high extraction productivity was achieved (97%-propionic acid), only a small portion vanished acid-amine complex formation. Pressure and temperature conditions need to be maintained as they have a high effect on productivity [213–215].

## 8. Advancing towards Sustainable Development of MES

### 8.1. Uses of Renewable Sources of Energy and Integrated Hybrid Systems

The best solution to the present-day challenges including resource scarcity, waste generation and sustaining economic benefits is an economy that is environmentally and economically regenerative like that of a circular bio-based economy depicted by MES [8,25,114]. Using alternative energy resources for the generation of biofuel are becoming an increasing trend. The idea of biorefinery has been suggested to encourage a bio-based economy to promote the use of renewable sources like biomass; to produce fuels, generate electricity, heat and other beneficial chemicals in a circular economy model stimulating material reusing and recycling [25,216]. This feedstock primarily comprises energy-generating crops and waste biomass, which is a readily available alternative that can partially substitute the current reliance on fossils providing a green source of feedstock for chemicals and fuel [217]. Renewable biomass can provide reliable, stable and sustainable energy, being easily accessible and continually replenished [218].

The decreased expense of facilities and installations for clean energy generation is a crucial factor in the ongoing development of the plant. Remarkable global investments in trending sustainable renewable technologies such as photovoltaics (PV), turbines that run on wind, hydro and biomass have technically made it easier and cost-effective to generate 1.9 to 6.3 fold more energy than the global energy demand from the renewable sources of biomass [219–222]. In addition, the finance division has provided low-interest rates on investment in clean energy [218]. As a result, energy needs to be retained and stored for a future supply-demand [223]. In addition, energy cannot be specifically cohered into chemical-based devices or fuel. Biomass is the only alternative green option, yet their production is limited because of lower efficiency [224]. Therefore, novel techniques that could directly turn electricity into fuel and chemicals are required. Hence, microbial electrosynthesis (MES) technology proves to be a promising solution [225].

Reducing carbon dioxide into fuels and multi-carbon organic chemicals has been described as an appealing method for the transformation of solar energy. However, non-biological electrochemical $CO_2$ removal is challenging [96]. The findings indicate that microbial catalysts could be a feasible solution and the current-driven microbial carbon dioxide reduction reveals an entirely different mode of photosynthesis when combined with PV [17]. Compared to traditional biomass-based methods, this turns solar energy more effectively into organic compounds. In this review paper, the authors have analysed and discussed the fundamentals of MES.

The bioelectrochemical processes require electrical energy. For instance, production of 1 kg of acetic acid consumed around an operational voltage of 3 V, and caproic acid requires double such power consumption. Without the cost of maintenance and DSP, the cost of electricity in producing 1 kg of acetic acids would be higher compared to its commercial value. The integration of energy from fossil fuels has various disadvantages in net generation and reduction of carbon emission. Therefore, the development of low-cost renewable energy resources is considered vital in the sustainable biorefineries development of MES. This development plan is less expensive and decreases the cost of electricity price by around 20 to 30% and is increasingly innovative compared to other sources, including energy based on fossil fuels. Renewable energy provides clean electric energy from natural recurrent sources like solar, wind, hydro, geothermal. Figure 5 depicts how these energy sources can be assimilated either indirectly to power MES or directly for product transformation using photoactive electrodes [8].

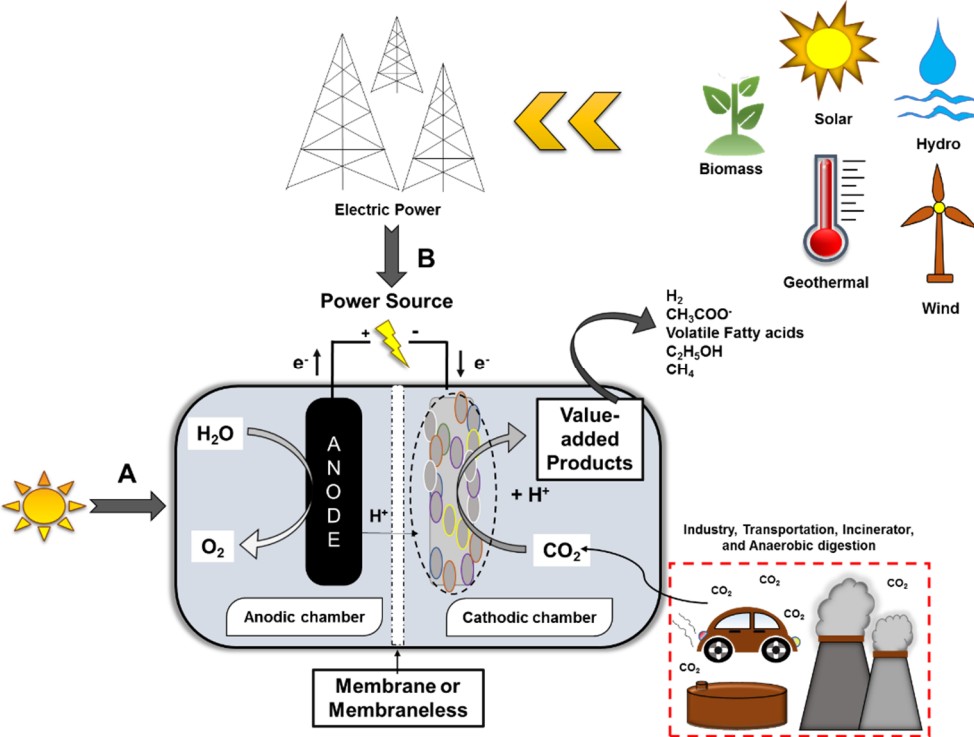

**Figure 5.** Microbial Electrosynthesis power supply with (**A**) Direct and (**B**) Indirect source. Production of high value-added chemicals and corresponding extracellular electron transfer (EET) processes interpreting the flow of electrons from the cathodic electrode to $CO_2$, the terminal electron acceptor [8].

Indirect renewable energy supply is considered more convenient as the excess power produced at a low cost by naturally fluctuating renewable sources can be stored as multi-carbon chemicals. Certain disadvantages of this strategy include the temporary decrease of MES production rate due to the fluctuating supply of electricity, and sometimes the metabolic pathway goes astray from carboxylic production to methane production. Exclu-

sive electronic circuits or batteries can be used to provide a constant current to the reactor which helps in avoiding these related issues in MES production.

The advantage of direct energy supply is that it is a self-supporting electricity source. This can be achieved by using light-emitting diodes like LEDs and a power storage unit for constant delivery of electric current to the cathode. Sufficient currents can be delivered using an advanced method like photo-electrochemical cells for water splitting to achieve $CO_2$ reduction electrochemically and also the production of methane, acetic acid and iso-propanol from $CO_2$ using photo-electrochemical anodes coupled with biological cathodes as portrayed in Table 6 below. Other than this, enzymatic biocathodes are also used in $CO_2$ reduction, but they cost higher and require periodic regeneration, making them less convenient for MES production [5,8].

**Table 6.** Hybrid system devices used in MES for wastewater treatment and $CO_2$ recycling. Adapted from [8].

| Inoculum | Cell Design | Cathode | Anode | Current (mA/cm$^2$) | Main Product (Yield/Final Concentration) | Coulombic Efficiency (%) | Solar Conversion Efficiency (%) |
|---|---|---|---|---|---|---|---|
| Engineered *Ralstonia eutropha* | Single chamber | NiMoZn or stainless steel | CoP$_i$ | 0.5–1.1 | Isopropanol (216 mg/L) | 3.9 | 0.7 |
| Enriched methanogenic community | Dual chambers | Carbon cloth | TiO$_2$ nanowire array | 0.07–0.09 | Methane (1.92 L/(m$^2$ d)) | 95 | 0.1 |
| *Sporomusa ovata* | Dual chamber | Si and TiO$_2$ nanowires arrays | TiO$_2$ nanowires | 0.3 | Acetic acid (6 g/L) | 86 | 0.38 |
| Effluent from methanogenic MES | Dual chamber | Chitosan modified carbon cloth | TiO$_2$/CdS on Fluorine-doped tin oxide (with copper zinc tin sulphide sensitiser) | 0.6 | Methane (15 L/(m$^2$ d), 20.8 L/(m$^2$ d) with copper zinc tin sulphide) | 93 | 0.62 (0.86 with copper zinc tin sulphide) |

Reducing the emission of $CO_2$ and wastewater treatment, both are challenges faced by industries. A combined process of oxidation and reduction at simultaneous electrodes, of pollutants and $CO_2$ respectively, are reported in solving these problems with the addition of decreasing the energy demand of MES. The reaction of oxygen evolution obtained in the process of wastewater oxidation at the anode and reduction of $CO_2$ at cathode require expensive and high potential catalysts, the oxygen produced at the anode inhibits the strictly anaerobic microorganisms by $O_2$ diffusing towards the cathode. The wastewater treatment at the anode produces carbon dioxides that reprocess at the cathode chamber and act as a precursor in the production of chemicals. Selective oxidation target compounds like alkene can be done by photocatalytic oxidation. The usefulness of this method is required to study further with the practical use of wastewater, but enhancement of both the anodic-cathodic reaction in combined systems can be challenging [195].

The sustainability analysis shown in Figure 6 was conducted for MES by which numerous crucial parameters such as production rates and energy usage can reduce the techno-economic and sustainable viability of biochemical production [17].

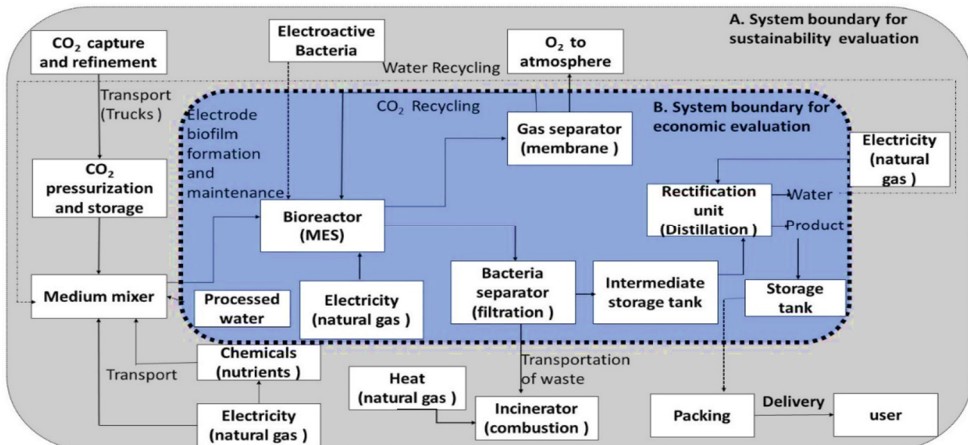

**Figure 6.** Schematic representation of sustainability and techno-economic assessment of MES.

### *8.2. Electronic Design and Energy Storage for MES*

As mentioned above MES requires a supply of electricity constantly for 24 by 7 functioning, but the reactor shows fluctuations of current flow. When there is a need for fluctuating energy sources to be used as a power source to MES, additional expenditure is required for the storage system for recovering the excessive energy for gaining and delivering that energy to MES reactors constantly. In the past 5 years, the cost price of energy storage systems like batteries have decreased by 60% and is expected to decrease further in the next 4 years duration. So installation of batteries for solar power harvesting is low at the cost, but other alternatives are available like, the energy storage devices that can be charged from the electric energy produced from wastewater treatment by MFC and this will eventually discharge to the power reactor of the MES system.

A change in a metabolic pathway or $CO_2$ recycling occurs when microbial cultures shift while mixed culture is being used, when there are non-uniform potentials between the stack of the cells, this can be created by an uneven distribution of charge on the MES electrode caused by the use of inhomogeneous microbial catalysts. The potential control in MES parallel stacks is difficult, this balancing can be achieved by a power management (PM) system present in the cell stacks. This PM system is used to switch off the connection by detecting the overloading and over-voltage occurring in the system, this reduces the stress on battery units and increases the lifespan, allowing stable chemical output in Despite the technological advances of the field, production rates far beyond the current record are required for MES commercialisation. Other similar switching systems are implemented to improve the output energy of MFC grids. Some of them include potentiostatic cell control which is expensive but effectively control electrode potentials and galvanostatic cell control, which is cheaper but shows fluctuation and division [226,227].

### *8.3. Commercialisation of MES*

Despite the technological advancements in this area, the productivity (both yield as well as cost) required for the massive scale preparation of MES is still beyond the current record, hampering its commercialisation. MES surely depicts a promising green future, but further studies, trials and researches are some of its prerequisites to accomplish the present need. Moreover, a reproducible, durable plan is needed for the carboxylate molecules' production, so far only acetic acid has been developed yet. Therefore, the paucity of knowledge must be resolved as a priority [7,17].

Industries such as dry cleaning, welding, preparation and processing of foaming agents and soft drinks utilise carbon dioxide [228]. Although, the amount of $CO_2$ used by these industries is negligible and does not significantly affect the levels of carbon emission in nature. International companies such as PRAXAIR are working on turning ambient carbon dioxide into a readily usable form by collecting and turning it into chemicals. Other companies working on the same subject include Novomer, Newlight and Algenol, which

turn atmospheric carbon dioxide into polypropylene carbonate, AirCarbon plastics and ethanol. Firms like Phycal are testing out unique approaches by opting for biological pathways in place of chemical pathways by culturing algae in ponds for $CO_2$ sequestration [152,229–231]. Further, the cultured algae are processed for producing oils and biofuels. Biofuels are advantageous since they do not require large masses of agricultural land or freshwater for cultivation, contrary to biomass-based fuel production. Table 7 describes various patents involved in microbial electrosynthesis technology.

**Table 7.** Several existing patents concerning microbial electrosynthesis (MES).

| Patent No. | Description | Ref. |
|---|---|---|
| US9856449B2 | The innovation offers mechanisms and approaches for the generation of various organic compounds by utilising $CO_2$ as an origin of carbon and electricity as an energy's point of origin. A reaction cell is supplied with an anode and a cathode (containing microbial biofilm) distinguished by the utilisation of a selectively porous membrane and conjugated to an electrical power source. The microbial biofilm contains an electrogen that can accept electrons and, in a cathode half-reaction, it can convert $CO_2$ to an organic compound and water, which is then decomposed into free molecular $O_2$ and $p^+$ in an anode half-reaction. The half-reactions are powered by electricity from an external source. Butanol, ethanol, formate, acetate and 2-oxobutyrate are the compounds that can be produced using this technology. | [232] |
| KR101892982B1 | As per the current innovation, a traditional carbon electrode surface has been altered with a positive amine compound to surge the amount of adsorption of an EAM biofilm on the surface while simultaneously, improving the efficiency of transfer of electrons by adding metal nanoparticles, thereby maximising the generation of several biofuels such as biomethanol and hexanol. | [233] |
| US10494596B2 | In specific, the system refers to MES, via which a microbial strain capable of collecting electrons from an electrode is used to generate $CO_2$, formate or $H_2$ in co-cultivation with a strain of microbial development such as methanogen, acetogen or other microbes capable of producing those products. | [234] |
| US20190301029A1 | A system of bioelectric processing of organic molecules like acetate is studied in this current disclosure. In addition, it also proposes strategies for generating a hydrocarbon-based product using $CO_2$ as the source of carbon. | [235] |
| WO2020053529A1 | The innovation discloses the process for the regeneration of the reactor's bioanode operation and the application of the reactor for the electrosynthesis of organic acids and organic waste alcohols. | [236] |
| CN111961691A | This innovation discloses the utilisation of a biocathode in MES for catalytic $CO_2$ reduction and synthesis of organic compounds. Preparation of the biocathode with *Ruminococcus*, *Clostridium* and *Lachnospiraceae* culture and injection of $CO_2$ into the cathode chamber, circulating aeration and setting the theoretical range for polarisation to be $-0.8$ V to 1.2 V (vs. Ag/AgCl). The invention reduces the $CO_2$ content drastically with a simultaneous high synthesising rate of organic compounds. | [237] |
| US10711318B2 | In comparison to the wild type of strain, a GM *Geobacter sulfurreducens* strain demonstrates enhanced functionality as a cathode biofilm. This strain is effective in utilising $CO_2$ as an origin of carbon and electricity as an origin of energy, employing a reverse tricarboxylic acid mechanism to produce a carbonaceous chemical. | [238] |
| CN110528017B | This paper demonstrates the bubbling tower of an electrolytic $H_2$ MES reactor. An electrolytic bath configured below the reactor supplies the bubble tower with micro-nano $H_2$ bubbles. $H_2$ and $CO_2$ are supplied by the microbes suspended in the bubble tower and then processed into organic compounds. This innovation is ideal for the method of $H_2$ Induced microbial $CO_2$ fixation, which is also relevant to the process of $H_2$-driven microbial sewage denitrification. This has perks of high coulombic performance, fast reactor start time, high current density, high output intensity, high system stability, compared to the conventional MES system dependent on the electroactive surface biofilm. | [239] |

The energy required in these systems is supplied from PV cells, which makes the system imitate the process of photosynthesis. The utilisation of carbon dioxide as a feedstock poses various challenges and drawbacks due to its inertness. Due to a low Gibbs energy value and relatively low or no reactivity these systems tend to demand more energy for

$CO_2$ sequestration into valuable products [186]. The biocatalysts used in MES not only make the system more efficient and cost-effective but also lower the energy needed to carry out the process. These microorganisms are functional even in mild environmental conditions, making the process more sustainable. Nevertheless, these biocatalysts can be a challenge when considering sensitivity and nutrient requirements at the field scale. Excess power and energy generated using renewable sources can later be stored using MES technology [240].

## 9. Prospects

The transition from traditional technologies to MES is a vital element of sustaining and protecting the ecosystem for the future and aiding the production of value-added commodities. Nevertheless, one cannot neglect that the parameters influencing the efficacy of such systems are yet to be optimised. When compared to other traditional technologies MES is facing serious issues with production yield. Another issue is in its electron transfer mechanism, which is a specifically crucial step for $CO_2$ reduction. Further study and involvement in topics such as the design, low-cost manufacturing and GM microbes and their metabolic pathways used in MES are therefore required to make MES technology suitable for large-scale implementation. To understand the potential improvements to make the device more successful, factors such as the pH, partial pressure, cultivation substrate, electrode potential, the architecture and feeding conditions of the reactor when optimised increase the efficiency of MES systems. Microbes present in the MES are also critical for the efficient functioning of the system; it should be noted that they are directly dependent on the mentioned parameters. The prime issue faced in MES and electro-fermentation is the variance in the $e^-$ transport mechanisms in the microbes, making it problematic to identify a universal model organism.

The scope of EET has increased in the past decade and has led to the utilisation of electric current with microbes impelling swift advancement in this field. The chief bottlenecks of electro-fermentation are the scarcity of available gene-editing tools to engineer metabolic pathways of target commodities, and electrode materials and operation of the BES reactors, which all together limits the implementation of electro-fermentation at scaled-up levels.

MES can also be employed for developing bio-refineries to store power in the form of organic compounds that can be used later. Hence, whatever energy is lost can be recovered and reused from the energy stored. Another bonus of MES technology is that it does not require agricultural land for the generation of biofuels. Future developments include the improvement of MES by using hybrid systems wherein MES systems are integrated with already established technologies such as an AD or a PV cell, increasing the bio-production process by about 9 folds. Therefore, such hybrid systems should be promoted and worked upon. Hence, using MES and optimising its parameters, more sustainable and environmentally friendly bio-refinery industries can be set up.

The trend of integrating several technologies is rapidly being accepted globally due to its resulting doubled advantages. MES can be integrated with other advanced technologies such as anaerobic fermentation, membrane electrolysis, $CO_2$ membrane separation, membrane contactors, microalgal photobioreactors and enzyme-assisted MES, to complement and improve the performance of $CO_2$ sequestration and make its conversion realistic and practicable.

## 10. Conclusions

- MES and Electro-fermentation are innovations that not only aim at minimising the emissions of greenhouse gases but also contributes to low manufacturing prices boosting the circular bioeconomy, offering a practical solution to lighten the ever-expanding global issues.
- Both these processes provide a plethora of premium products like biofuels, bioenergy and can also perform concurrent valorisation of $CO_2$ and wastewater.

- Recently, there have been multiple strategies in optimising the MES process and improving its efficiency, including treating the cultivation substrate to include adequate nutrients, enhancing the architecture and feeding conditions of the reactor, enriching the inoculum mix culture, running reactors in optimised conditions and also boosting the microbial interactions by spatially organising the cathode.
- However, significant challenges need to be tackled before commercialisation. Both these technologies, in general, are still far from practical application and further research into basic operational variables, long-term stability, continuous production, modelling, repeatability and scalability is still necessary.
- Overall, this review paper promotes further studies on promising microbial aspirants to aid advancement in this emergent field, with the subsequent aim of bringing this sustainable technology one step closer to real-world applications.

**Author Contributions:** Conceptualisation, S.P., D.L. and D.A.J.; software, K.W. and M.Q.; validation, S.P., R.R.R. and R.P.; writing—original draft preparation, M.Q., K.W. and S.P.; writing—review and editing, S.P., M.Q., D.A.J., A.K.R. and P.K.G.; supervision, S.P.J., V.K.T., R.R.R. and R.P. All authors have read and agreed to the published version of the manuscript.

**Funding:** Authors duly acknowledge the grant received from Sharda University seed grant project (SUSF2001/01).

**Institutional Review Board Statement:** Not applicable.

**Informed Consent Statement:** Not applicable.

**Data Availability Statement:** Not applicable.

**Conflicts of Interest:** The authors declare no conflict of interest.

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
