# Peer review of "Valorisation of CO2 into Value-Added Products via Microbial Electrosynthesis (MES) and Electro-Fermentation Technology"

_fermentation, doi:10.3390/fermentation7040291_

Round 1

Reviewer 1 Report

At the very first glance, I thought the authors did a throughout revision on the paper, but after a closer look, the changes were more on the level of structure: shuffling the paragraphs, renaming the subheadings, etc. I was already disappointed by reading the first page that the authors were still binding the MES concept for energy production (line 30-31), which was obviously not correct. The product from MES can be used as precursors for biofuel production, but one cannot say the purpose of MES is to produce biofuel. The scientific logic is not correct. These are similar in many of the comments I had for the previous submission. However, the authors did not get it correctly and now the new version is still not convincing.

More importantly, I am highly concerning that I might detect a potential plagiarism in this paper. The table 1 in this paper shows too much similarity with the Table 1 from the paper below (https://doi.org/10.1016/j.synbio.2020.08.004.), especially for the text in the column of Genetic modification of host and Yield. It will be really hard for me to believe the authors made the table by themselves independently. 

Thus, I will still highly suggest a reject to this paper, and hope the authors can take my comments more seriously and also do proper citing if they "use" some contents from other papers. 

Author Response

Response to reviewers' comments

Manuscript ID: Fermentation-1414447

Valorization of CO2 into value-added products via Microbial Electrosynthesis (MES) and Electro-fermentation technology

The authors would like to appreciate all the reviewers for their valuable suggestions and comments involved in enhancing the standard of our manuscript.

 Reviewer #1:

 At the very first glance, I thought the authors did a throughout revision on the paper, but after a closer look, the changes were more on the level of structure: shuffling the paragraphs, renaming the subheadings, etc. I was already disappointed by reading the first page that the authors were still binding the MES concept for energy production (lines 30-31), which was not correct. The product from MES can be used as precursors for biofuel production, but one cannot say the purpose of MES is to produce biofuel. The scientific logic is not correct. These are similar in many of the comments I had for the previous submission. However, the authors did not get it correctly and now the new version is still not convincing.

More importantly, I am highly concerned that I might detect potential plagiarism in this paper. Table 1 in this paper shows too much similarity with Table 1 from the paper below (https://doi.org/10.1016/j.synbio.2020.08.004), especially for the text in the column of Genetic modification of host and Yield. It will be really hard for me to believe the authors made the table by themselves independently.

Thus, I will still highly suggest a rejection to this paper and hope the authors can take my comments more seriously and also do proper citing if they "use" some contents from other papers.

 First of all, the authors would like to appreciate the reviewer for pointing it out again, we agree that it is a valid point. We understand now what the reviewer was trying to say in the earlier comments. We have made the required changes throughout the revised manuscript. As suggested, we have made the necessary changes in Table 1 to reduce the plagiarism and the proper citation has been added in the revised manuscript.

Reviewer #2:

 This review accentuates the concept, importance, and opportunities of MES, as an emerging discipline at the nexus of microbiology and electrochemistry. Production of organic compounds from MES is considered as an effective technique for the generation of various beneficial reduced end-products as well as in reducing the load of CO2 from the atmosphere to mitigate the harmful effect of greenhouse gases in global warming. The authors have focused on MES, as it is the next transformative, viable alternative technology to decrease the repercussions of surplus carbon dioxide in the environment along with conserving energy. The review is interesting and covers many aspects of the subject. The paper is of a satisfactory level, quite clear and well organized with strong discussion.

Reviewer #3:

 As a current review, this article is decently developed, with discoveries mostly cited from recently published studies on microbial electrosynthesis systems. In my opinion, this article is ready for publication after a few edits. My specific comments/suggestions are as follows.

R3Q1: Keywords: Some keywords are too broad and repetitive, and should be replaced/removed. For example, electrochemical cell, extracellular electron transforms, downstream processing, electrobiotechnology, integrated approaches. The keywords must be directly relevant to and focused on this article.

R3A1: As recommended, we have removed the repetitive keywords and instead added the ones directly relevant to this manuscript.

R3Q2: Figure 2: According to its title, this article is supposed to focus on synthesizing value-added products using CO2 as electron acceptors. This microbial fuel cell (MFC) example here, which uses proton as an electron acceptor, can be misleading and do not match the topic. I suggest replacing this example and its relevant discussion.

R3A2: We want to explain Microbial Fuel Cell as a primary invention/tool of BES. However, as per suggestion from reviwer, we have included a typic microbial electrosyntheis where CO2 can be sequested to produce value added product. The microbial electrosyntheis is an adapted version of MFC. Unlike anode in MFC, cathode of the microbial electrosyntheis is involved in electron transfer and responsible for the reduction of CO2 to value added chemicals.

R3Q3: Other examples in this article should also serve for the topic specified in the title. I also suggest replacing some examples in Table 1.

R3A3: We have accordingly made changes in the mentioned table in the revised manuscript.

R3Q4: Section 8. 3: The challenges regarding the commercialization of MES should be elaborated.

R3A4: As suggested, we have added the main aspects that hinder the commercialization of MES technology, “In spite of the technological advancements in this area, ….be resolved as a priority”, in the revised manuscript.

R3Q5: Conclusions: These can be more concise.

R3A5: We have further made this section concise in the revised manuscript.

R3Q6: Thoroughly check grammatical errors.

R3A6: We have thoroughly checked for grammatical errors throughout the revised manuscript.

Reviewer 2 Report

This review accentuates the concept, importance, and opportunities of MES, as an emerging discipline at the nexus of microbiology and electrochemistry. Production of organic compounds from MES is considered as an effective technique for the generation of various beneficial reduced end-products as well as in reducing the load of CO2 from the atmosphere to mitigate the harmful effect of greenhouse gases in global warming. The authors have focused on MES, as it is the next transformative, viable alternative technology to decrease the repercussions of surplus carbon dioxide in the environment along with conserving energy. The review is interesting and covers many aspects of the subject. The paper is of a satisfactory level, quite clear and well organized with strong discussion.

Author Response

(The authors gave the same response as above.)

Reviewer 3 Report

As a current review, this article is decently developed, with discoveries mostly cited from recently published studies on microbial electro synthesis systems. In my opinion, this article is ready for publication after a few edits. My specific comments/suggestions are as follows.

1) Some keywords are too broad and repetitive, and should be replaced/removed. For example, electrochemical cell, extracellular electron transform, downstream processing, electrobiotechnology, integrated approaches. The keywords must be directly relevant to and focused on this article.

2) Figure 2: according to its title, this article is supposed to focus on synthesizing value-added products using CO2 as electron acceptors. This microbial fuel cell (MFC) example here, which uses proton as electron acceptor, can be misleading and do not match the topic. I suggest replacing this example and its relevant discussion.

3) Other examples in this article should also serve for the topic specified in the title. I also suggest replacing some examples in Table 1.

4) The challenges regarding the commercialization of MES should be elaborated in Section 8.

5) Conclusions can be more concise.

6) Thoroughly check grammatical errors.

Author Response

(The authors gave the same response as above.)
